# Out of equilibrium mean field dynamics in the transverse field Ising model

I. Homrighausen[1], S. Kehrein[1*],

**1** Universität Göttingen, Institute for Theoretical Physics,
Friedrich-Hund-Platz 1, 37077 Göttingen, Germany
* stefan.kehrein@theorie.physik.uni-goettingen.de

October 24, 2019

## Abstract

We investigate the quench dynamics of the transverse field Ising model on a finite fully connected lattice. Using a rate function approach we compute the leading order corrections to the mean field behavior analytically. Our focus is threefold: i) We analyze the validity of the mean field approximation and observe that deviations can occur quickly even for large systems. ii) We study the variance of the order parameter and identify four dynamically qualitative different regions. iii) We derive the entanglement Hamiltonian for a bipartition of the lattice, which turns out to be a time-dependent harmonic oscillator.

# 1   Introduction

One of the reasons why quantum mechanical many body systems are difficult to analyze is because the dimension of the Hilbert space grows exponentially with the number of particles. In contrast, the dimension of classical phase space scales only linear in the particle number. Another unique feature of quantum mechanics is entanglement, which has no immediate classical analog [1, 2]. When entanglement of a composite system is measured by means of the von Neumann entanglement entropy, the logarithm of the Hilbert space dimension of the smaller subsystem is an upper bound on the entanglement. Turning this intuition around, one can view the exponential of the entanglement entropy as the effective dimension in which the entangled state lives. In this sense, the combination of both, large entanglement and exponential Hilbert space dimension, makes the quantum time evolution computationally challenging. Many numerical algorithms, such as the density matrix renormalization group [3–5] with matrix product states [6], rely on the fact that the entanglement of the states of interest remains low such that the effective Hilbert space dimension is small and the complexity of the exponential dimension is effectively avoided. Generic quantum many body systems are not exactly solvable and one is restricted to numerical methods and finite computational resources. From this perspective, it is vital to understand how entanglement grows in time in non-equilibrium situations.

The entanglement dynamics after a global quantum quench has been investigated for numerous local Hamiltonians. Linear growth of the entanglement entropy has been observed

for one dimensional gapped lattice systems [7], conformal field theories [8,9], non integrable spin chains [10], and harmonic oscillator chains [11]. This typically limits the study to low dimensional locally interacting systems, for which the area law [12–16] guarantees low entanglement entropy in the ground state, and to small system sizes at early times. The linear growth of entanglement as discussed in [7–9,17], is mediated by quasiparticles propagating in a Lieb-Robinson cone formed by a maximal group velocity. The quasiparticle picture has been confirmed analytically in integrable models [8,9], as well as numerically, e.g. by looking at the mutual information between two spatially separated places [18], or the particle number fluctuation [19]. There are, however, exceptions to the connection between entanglement growth and the spread of quasiparticles. On the one hand, it is known [10] that some non integrable models show linear entanglement growth, while the energy transport, being mediated by quasiparticles, is only diffusive. On the other hand, sublinear entanglement growth was observed in geometric quenches, even though quasiparticles spread ballistically [20].

A notable exception to linear entanglement growth in short range systems are disordered models that exhibit many body localization and show logarithmic entanglement growth [21–23]. The logarithmic growth can be argued to be a consequence of a dephasing mechanism facilitated by exponentially decaying interactions between localized quasiparticles [23].

In addition to short range models, systems with long range interaction have gained theoretical [24–33], as well as experimental interest due to their realization with ultra cold atoms [34,35]. Another, more theoretical, motivation to study long range models is to use them as an approximate equivalent for high dimensional short range systems [19]. It has been found numerically [18,36,37] and semi-analytically [27,28] that the entanglement entropy grows only logarithmically in time, that is much slower than their short range counterpart. A heuristic, non-quantitative argument in favor of the logarithmic growth [27], also see [19], relies on the fact that the maximal group velocity diverges for the $k = 0$ mode, while the density of states vanishes as $k \to 0$. This leads to a breakdown of a pronounced light cone, and information is only propagated slowly by quasiparticles. However, this line of reasoning cannot be applied to the limiting case of uniform all to all coupling, because fully connected models lack the notion of spatial distance and a quasiparticle picture.

In the present paper, we look at the out of equilibrium dynamics in an infinite range, highly symmetric model, which becomes amenable to a mathematically controlled expansion in the thermodynamic limit. More precisely, we focus on a spin system defined on a fully connected lattice, being invariant under permutations of lattice sites. For the sake of concreteness, we will focus on the fully connected transverse field Ising model (also known as the Lipkin Meshkov Glick model [38]), however, the mathematical reasoning also applies to other mean-field models on fully connected lattices [39].

Mean field models, and mean field approximations of more complicated systems provide an accessible approach to study many body problems, both, in classical, and quantum statistical physics. The applications of mean field approximations in equilibrium situations are numerous, and it is rather well understood when mean field yields reliable results. In contrast, mean field approximations are less frequently used in non equilibrium conditions, and it is not generally known when and how well mean field works. From this point of view, the transverse field Ising model serves as a basic and non-trivial example to study the validity of approximations out of equilibrium. Two advantages of this specific model are that, first, it is accessible to controlled analytical calculations, and, second, because numerically exact solutions for large system sizes are feasible, it is possible to compare the approximations to exact results. One of the surprising findings is how short the time scale of validity of the mean field

approximation in this system is. More specifically, we show that, away from critical points, mean field is only reliable for early times of the order of the square root of the system size. And, close to unstable critical points, the mean field approximation already breaks down on timescales logarithmic in system size.

When driving the fully connected Ising model out of equilibrium by means of a sudden quantum quench, the dynamics is constrained to the site permutation invariant subspace, which is referred to as the Dicke subspace. The dimension of the Dicke subspace scales linearly with the number of spins, which reminds of the scaling of classical phase spaces. Indeed, permutation invariance facilitates the use of semiclassical techniques. In this way, the quench dynamics in the transverse field Ising model on a fully connected lattice becomes amenable to a mathematically controlled expansion around the classical limit, and is a useful test case to benchmark the validity of mean field type approximations out of equilibrium.

Spin systems on fully connected lattice geometries can be viewed as a single collective spin, and are thus mathematically equivalent to the two mode Bose Hubbard model [40, 41] via the Jordan-Schwinger mapping [42, 43]. The two mode Bose Hubbard model is experimentally realized as a Bose-Einstein condensate (BEC) using ultra cold atoms in optical traps [34, 35]. In this context, entanglement between the two modes has been investigated theoretically [29–32] and experimentally [35, 44]. A typical entanglement measure between the modes of a dimer is referred to as EPR-entanglement. Besides the entanglement between the two modes of a BEC dimer, one may also investigate the entanglement between different particles of the BEC, which corresponds to a different bipartition of the Hilbert space [30]. In this paper, we focus on the entanglement between particles.

Although being a relatively simple model, the entanglement dynamics in the mean field Ising model is non-trivial and exhibits qualitatively different behavior, such as linear growth, logarithmic growth, and bounded oscillations, depending on the initial pre-quench state and the final post-quench Hamiltonian. Remarkably, within the validity of the mean-field approximation we can analytically derive the complete entanglement Hamiltonian in leading order, which turns out to be a time-dependent harmonic oscillator. This provides a rare case where the complete entanglement Hamiltonian and therefore all Rényi entanglement entropies are analytically known for a non-trivial quantum many body system. The dynamical behavior can be understood by making use of an intimate connection between entanglement and spin squeezing [33, 40, 45–49].

Throughout the paper, we compare analytical predictions to numerical data obtained by exact diagonalization, and find excellent agreement at early times. The fact that the Dicke subspace dimension scales linearly with the number of spins, allows one to solve systems of $10^4$ spins numerically exact. However, even for large system sizes a dephasing mechanism leads to a deviation from the mean field approximation as time proceeds.

This article is structured as follows. The fully connected transverse field Ising model is defined in Sec. 2, and the mapping to an effective semiclassical model in the limit of large system size is explained. In Sec. 3, two semiclassical techniques, one based on a rate function expansion, the other based on deviations between classical trajectories, is reviewed. These techniques are used in the discussion of the quench-induced dynamics of the mean magnetization and its variance, see Sec. 4, and the entanglement entropy with respect to a bipartition of spins, see Sec. 5. The dynamical phase diagram based on the behavior of the order parameter and the variance is discussed in Sec. 4 and entanglement is analyzed in Sec. 5. The article concludes with Sec. 6.

## 2  Mean field models

### 2.1  Transverse field Ising model

We investigate the transverse field Ising model on a fully connected graph of $N$ sites given by the Hamiltonian

$$\mathcal{H} = -\frac{J}{2N} \sum_{i,j} s_i^z s_j^z - \Gamma \sum_i s_i^x, \tag{1}$$

where $s_i^{x,y,z} = \sigma_i^{x,y,z}/2$ denotes the spin $1/2$ representation of the spin at site $i$ in terms of the Pauli matrices, $\Gamma$ is the homogeneous transverse field, and $J > 0$ denotes the ferromagnetic coupling. Note that the double sum is rescaled by a factor of $1/N$ in order to make it of the same order of magnitude as the single sum. In this way, both terms, the ferromagnetic term and the transverse term, scale linear with the system size such that the Hamiltonian is extensive. The linear scaling becomes more apparent when introducing the (rescaled) total spin operators $S_{x,y,z} = \sum_i s_i^{x,y,z}/N$ in terms of which the Hamiltonian (1) reads

$$\mathcal{H} = -NJS_z^2/2 - N\Gamma S_x.$$

The factor of $1/N$ in the definition of $S_{x,y,z}$ is chosen such that its spectrum consists of $(N+1)$ equidistant points contained in the interval $[-1/2, 1/2]$. One can thus view $S_{x,y,z}$ as a quantity of order one as $N \to \infty$. Note that $S_{x,y,z}$ obey the usual $SU(2)$ commutation relations decorated with an additional factor of $\hbar_{\text{eff}} := 1/N$. In the sequel, we choose units of time and energy in which $\hbar = 1$ and $J = 1$.

### 2.2  Dicke subspace and effective Hamiltonian

The Hamiltonian (1) is defined on the Hilbert space $\mathcal{H}_N = \bigotimes^N \mathbb{C}^2$. The tensor products $|s_1, \ldots, s_N\rangle$ of the $s_i^z$ eigenstates $|s_i = \pm 1/2\rangle$ form an orthonormal basis of $\mathcal{H}_N$. An important subspace of $\mathcal{H}_N$ is the Dicke space $\mathcal{D}_N$ containing all states that are invariant under permutations of spins. A convenient orthonormal basis of $\mathcal{D}_N$ is given by the Dicke states $\{|N_+\rangle\}_{N_+=0,\ldots,N}$, being defined as the superposition of all spin permutations with exactly $N_+$ of $N$ spins being up,

$$|N_+\rangle = \binom{N}{N_+}^{1/2} P\left(|\uparrow\rangle^{\otimes N_+} \otimes |\downarrow\rangle^{\otimes N-N_+}\right),$$

where $P$ denotes the projection operator $P|s_1, \ldots s_N\rangle = \frac{1}{N!} \sum_{p \in \mathcal{S}_N} |s_{p(1)} \ldots s_{p(N)}\rangle$ and $S_N$ denotes the symmetric group on $N$ symbols. The Dicke state $|N_+\rangle$ is the permutation invariant eigenstate of $S_z$ with eigenvalue $(n_+ - 1/2)$. Note that $\mathcal{D}_N$ is $(N+1)$ dimensional, i.e. its dimension scales linearly with the system size, as opposed to the exponential scaling of the $2^N$ dimensional total Hilbert space $\mathcal{H}_N$. The fact that the dimension of $\mathcal{D}_N$ scales only linear in $N$ allows to study the dynamics using exact diagonalization for large systems of the order of $N = 10^4$.

In this paper we study the non-equilibrium dynamics after a sudden quantum quench $\Gamma_i \to \Gamma_f$ in the magnetic field. That is to say, the system is prepared in the ground state $|\Psi_0\rangle$ of the pre-quench Hamiltonian $\mathcal{H}(\Gamma_i)$ and is evolved with the post-quench Hamiltonian $\mathcal{H}(\Gamma_f)$ according to the Schrödinger equation. On a fully connected lattice, both, the Hamiltonian (1) as well as the ground state, are invariant under spin permutations. Hence, in a quench

setup, the dynamics is confined to $\mathcal{D}_N$ and the wave function can be expanded in terms of the Dicke states as

$$|\psi\rangle = \sum_{N_+=0}^{N} \psi(n_+)|N_+\rangle$$

($n_+$ being $N_+/N$).

The time dependent Schrödinger equation $i\partial_t|\Psi\rangle = \mathcal{H}|\Psi\rangle$ imposes the dynamics

$$i\hbar_{\text{eff}}\partial_t\psi(n_+) = H(n_+,p)\,\psi(n_+) \tag{2a}$$

on the coefficients $\psi(n_+) = \langle N_+|\psi\rangle$ with the effective Hamiltonian

$$H(n_+,p) = -\frac{1}{2}\,(n_+ - 1/2)^2 - \Gamma\sqrt{n_+ - n_+^2}\,\cos(p), \tag{2b}$$

where $p = -i\hbar_{\text{eff}}\partial_{n_+}$ and $\hbar_{\text{eff}} = 1/N$. Details on the derivation of the effective Hamiltonian are given in Appendix A and [39], also see [50–52] for a derivation in the context of Bose-Einstein condensate dimers starting from a Gross-Pitaevski description. The effective description by Eq. (2) is an approximation because of two reasons. First, additional terms in $H(n_+,p)$ that are suppressed by $1/N$ are neglected. Second, the discrete nature of $n_+$ (taking values in $\{0, 1/N, \dots 1\}$) is approximated by treating $n_+$ as a continuous variable with values in the unit interval $[0,1]$. These approximations are believed to be valid as $N \to \infty$. Equation (2) has the form of an effective one dimensional single particle Schrödinger equation for a fictitious particle. The position of the fictitious particle is given by the fraction $n_+ = N_+/N$ of up-spins, and the conjugate momentum $p = -i\hbar_{\text{eff}}\partial_{n_+}$ can be interpreted as the polar angle on the Bloch sphere. As the effective Planck constant $\hbar_{\text{eff}}$ is the inverse system size, we may exploit semiclassical techniques in the large system limit to investigate the non-equilibrium dynamics after a sudden quench.

## 3   Semiclassics

Two semiclassical methods are presented. First, in the subsequent section, a systematic rate function expansion akin to WKB theory is discussed. This method gives a systematic $1/N$-expansion of the expectation value and the variance of observables and their dynamics. The main result will be Eq. (7), which is a simple ordinary differential equation describing the dynamics of the leading contribution to the variance. Second, thereafter in section 3.2, a semiclassical phase space approach, known as nearby orbit approximation [53,54], is reviewed. This method is particularly suited to facilitate an intuitive way of thinking and complements the less intuitive rate function expansion. We will take great advantage of this phase space picture when we explain the periodically enhanced spin squeezing. Both methods, the rate function expansion and the nearby orbit approximation, give identical results for the leading order term of the variance. This equivalence is proved in Appendix C.

### 3.1   Rate function expansion

In the large $N$ limit the ground state of (2b) may be approximated by WKB-type states [55–57] of large deviation form

$$\psi(n_+) \asymp e^{-Nf(n_+)} \tag{3}$$

with $N$-independent complex rate function $f(n_+)$ [39]. Following the notation of [58, 59], we write $a \asymp b$ to denote that two quantities are equal to first order in their exponents, i.e. $\lim_{N \to \infty} \frac{1}{N} \log a/b = 0$. The modulus of $\psi(n_+)$ is localized around the minimum of $\Re f(n_+)$. We assume that $\Re f(n_+)$ has a unique global minimum denoted by $n_{\mathrm{cl}}$. The expectation values $\langle n_+ \rangle$ and $\langle p \rangle$ in the state (3) follow from a leading order saddle point approximation to be

$$\langle n_+ \rangle = n_{\mathrm{cl}} + \mathcal{O}(1/N), \tag{4a}$$

$$\langle p \rangle = p_{\mathrm{cl}} + \mathcal{O}(1/N), \tag{4b}$$

where $p_{\mathrm{cl}} = if'(n_{\mathrm{cl}})$. Moreover, the curvature of the rate function at $n_{\mathrm{cl}}$ determines the variance $\mathrm{var}(n_+) = \langle (n_+ - \langle n_+ \rangle)^2 \rangle$ and $\mathrm{var}(p) = \langle (p - \langle p \rangle)^2 \rangle$. If we denote the second derivative $f''(n_{\mathrm{cl}})$ by $f_2$, we have

$$\mathrm{var}(n_+) = \frac{1}{2N} (\Re f_2)^{-1} + \mathcal{O}(1/N^2), \tag{5a}$$

$$\mathrm{var}(p) = \frac{1}{2N} \left[ \Re(f_2^{-1}) \right]^{-1} + \mathcal{O}(1/N^2). \tag{5b}$$

Likewise, all higher moments may be computed systematically in this perturbative manner by the saddle point approximation.

Now, we investigate the time evolution of the expectation value and its variance to leading order in $1/N$. In order to avoid ordering ambiguities, we assume that the Hamiltonian $H(n_+, p)$ in (2) is normal ordered in the sense that the momentum operator $p$ is commuted to the right. Then, the effective Schrödinger equation (2) imposes the partial differential equation

$$\partial_t f(n_+, t) = iH(n_+, i\partial_{n_+} f(n_+, t)) + \mathcal{O}(1/N) \tag{6}$$

on the rate function. Consequently, the quantities $n_{\mathrm{cl}}$, $p_{\mathrm{cl}}$, and $f_2$ become time dependent. As was shown in [39] $n_{\mathrm{cl}}(t)$ and $p_{\mathrm{cl}}(t)$ obey the classical Hamiltonian equations with Hamiltonian $H$. Elaborating on this result, we derive the differential equation

$$i\frac{df_2}{dt} = -(1, if_2)H''(1, if_2), \tag{7}$$

for $f_2$, where $H''$ is the two by two Hessian matrix of $H(n_+, p)$ evaluated at $n_+ = n_{\mathrm{cl}}$, $p = p_{\mathrm{cl}}$ in Appendix B. The time-dependence of $f_2$ yields the dynamics of $\mathrm{var}(n_+)$ and $\mathrm{var}(p)$ according to Eq. (5). It is a non-trivial fact that the time evolution of the variances does not depend on higher moments, such as the skewness, but only on the expectation values. This is a special case of a more general result. Namely, that the dynamics of the leading order of the $n$th moment depend only on moments of order smaller than $n$ (more details in Appendix B).

## 3.2 Phase space picture

The preceding paragraph introduced a systematic large $N$ expansion of the rate function. The computation of the variance is reduced to the solution of the ordinary differential equation (7) of the rate function's curvature at the classical trajectory. In the present paragraph we introduce a complementary semiclassical technique, which is based on a phase space picture.

The idea of a phase space formulation of quantum mechanics has a long-standing history and goes back to Wigner and Moyal [60, 61]. In a nutshell, phase space methods map the quantum mechanical wave function to a quasi-probability distribution on phase space whose

dynamics is then inherited from the Schrödinger equation [62–64]. One of the most commonly used quasi-probability distribution is the Wigner function and its evolution is governed by Moyal's equation. Operator expectation values are then obtained by integrating the Weyl symbol of that operator against the Wigner function over the whole phase space.

The leading contribution as $\hbar_{\mathrm{eff}} \to 0$ of the Moyal equation is the classical Liouville equation. Corrections to the Liouville's equation are suppressed by at least $\hbar_{\mathrm{eff}}^2$ [60]. As we are only interested in the leading order dynamics as $1/N \to 0$, we will approximate the Moyal equation by Liouville's equation. This is sometimes referred to as the truncated Wigner approximation and it is exact for quadratic Hamiltonians. As innocent as this approximation seems, it is known that the limit $\hbar_{\mathrm{eff}} \to 0$ may have an essential singularity and the truncated Wigner approximation may be insufficient in this case [65]. This issue, however, is less relevant for us, as we consider only those quenches, for which the initial Wigner function can be approximated by a single Gaussian. The mean of this initial Gaussian is given by $(n_{\mathrm{cl}}(0), p_{\mathrm{cl}}(0))$ (compare Eq. (4)), and the covariance matrix $C(0)$ is diagonal with eigenvalues $[\Re(2Nf_2(0))]^{-1}$ and $\Re[(2Nf_2(0))^{-1}]$ (compare Eq. (5)). As the initial Wigner function is strongly localized, on a scale of $1/\sqrt{N}$ in phase space, the nearby orbit approximation [53, 54] predicts that the evolved Wigner function at a later time $t$ can be approximated by a Gaussian distribution centered at the classical reference orbit passing through $(n_{\mathrm{cl}}(0), p_{\mathrm{cl}}(0))$ with covariance

$$C(t) = S(t)C(0)S(t)^T. \tag{8}$$

Here $S(t)$ is the linear approximation, i.e. the Jacobian matrix, of the classical Hamiltonian flow and is thus a symplectic two by two matrix (see also Appendix D for further details). In other words, $S(t)$ is the fundamental solution of Hamilton's equations of motion linearized around the reference orbit and obeys the non-autonomous differential equation

$$\dot{S}(t) = JH''(n_{\mathrm{cl}}(t), p_{\mathrm{cl}}(t))S(t) \tag{9}$$

with initial condition $S(0) = \mathrm{id}$. The nearby orbit approximation is due to Heller et al. and Littlejohn et al. [66–70] and was further developed e.g. in [71–74] (also see [53,54] for extensive reviews). A related, though different approximation is discussed in [75].

We stress that Eq. (8) involves two approximations. First, the full quantum dynamics is approximated by the classical Liouville equation of the Wigner function. And second, as the initial Wigner function is a strongly localized Gaussian in phase space, Liouville's equation is approximately solved by a Gaussian centered at the classical reference orbit within the nearby orbit approximation. As shown in Appendix C, Eqs. (7) and (8) are equivalent.

## 4 Results for the variance

We now discuss the dynamics of the expectation value of spin operators in the spin system (1) after a sudden quantum quench in the external magnetic field $\Gamma$. More specifically, we are interested in the dynamics of the magnetization per site $\langle n_+ \rangle$ and its variance. That is, we prepare the initial state as the ground state of the pre-quench Hamiltonian $\mathcal{H}(\Gamma_i)$ with an external magnetic field $\Gamma_i$ and evolve the state with the post-quench Hamiltonian $\mathcal{H}(\Gamma_f)$, where $\Gamma_f$ is different from $\Gamma_i$ such that the post-quench Hamiltonian does not commute with the pre-quench Hamiltonian and the dynamics is non-trivial.

Preparing the state in the ground state of the pre-quench Hamiltonian fixes the initial conditions $n_{cl}(0)$, $p_{cl}(0)$, and $f_2(0)$. As before, we denote the global minimum of $\Re f$ by $n_{cl}$, and write $p_{cl}$ for $if'(n_{cl})$ and $f_n$ for the Taylor coefficients $f^{(n)}(n_{cl})$. The ground state of the pre-quench Hamiltonian obeys the eigenvalue equation $H(n_+, -i\partial_{n_+}/N)e^{-Nf(n_+)} = Ee^{-Nf(n_+)}$ with ground state energy $E$. By neglecting zero-point fluctuations in the energy $E$, which are of order $1/N$, we may write

$$H(n_+, if') \approx E \tag{10}$$

instead of $[H(n_+, if') + \mathcal{O}(1/N)]e^{-Nf(n_+)} = Ee^{-Nf(n_+)}$. Taking the first derivative of (10) w.r.t. $n_+$ at $n_{cl}$, yields $H^{(1,0)} + if_2 H^{(0,1)} = 0$, which is solved by any critical point $(n_{cl}, p_{cl})$ of $H$. Since we are interested in the ground state, we choose the absolute minimum (assuming it exists and is unique). Intuitively, the ground state Wigner function is only significantly different from zero in the neighborhood of the absolute minimum of $H$, which gives the main contribution to $E$. The fact that the Wigner function is only localized on a scale of $1/\sqrt{N}$ in phase space, leads to additional (zero-point) contributions of order $1/N$ to the energy. Taking the second derivative of (10) at $n_+ = n_{cl}$ yields $H^{(2,0)} + 2if_2 H^{(1,1)} + (if_2)^2 H^{(0,2)} + if_3 H^{(0,1)} = 0$. Using $H^{(0,1)} = 0$, the last equation can be written in matrix form as $(1, if_2)H''(1, if_2) = 0$, where $H''$ denotes the Hessian matrix evaluated at the critical point $(n_{cl}, p_{cl})$. This quadratic equation in $f_2$ and can be readily solved.

The initial condition are thus determined by

$$n_{cl}(0) = \begin{cases} (\pm\sqrt{1 - 4\Gamma_i^2} + 1)/2, & \Gamma_i < 1/2 \\ 1/2, & \Gamma_i > 1/2, \end{cases} \tag{11a}$$

$$p_{cl}(0) = 0, \tag{11b}$$

$$if_2(0) = -\frac{H^{(1,1)}}{H^{0,2}} \pm \sqrt{\frac{H^{(1,1)}}{H^{0,2}} - \frac{H^{(2,0)}}{H^{(0,2)}}} \tag{11c}$$

($H^{(n,m)}$ being the $n$th and $m$th derivative of $H$ w.r.t. its first and second argument, respectively, evaluated at $(n_{cl}, p_{cl})$). Equations (11a) and (11b) determine the absolute minimum of the Hamiltonian function $H$ [39]. The critical points of $H(n_+, 0)$ undergo a pitchfork bifurcation at the critical point $\Gamma_c = 1/2$. In the ferromagnetic phase, for $\Gamma_i < \Gamma_c$, the symmetry under spin-flips leads to the two-fold degeneracy of the ground state in the thermodynamic limit. This is reflected by the fact that $H$ has two minima on equal footing. From now on, we tacitly assume that the spin-flip symmetry is broken, e.g. by adding the infinitesimal longitudinal field term $\epsilon S_z$ with $\epsilon = \mathcal{O}(1/N)$ to the Hamiltonian (1), and thereby singling out the positive square root in (11a). Note, that (11a)-(11c) is a fixed point of the classical equations of motion and Eq. (7) for $\Gamma = \Gamma_i$, i.e. when no quench is done. However, for $\Gamma = \Gamma_f \neq \Gamma_i$ the dynamics is non-trivial (see Figs. 2 and 3).

Figure 1 depicts six particular qualitatively different quenches in a dynamical phase diagram as pairs of $(\Gamma_i, \Gamma_f)$. This dynamical phase diagram was discussed by Biroli and Sciolla in [39] in the context of dynamical phase transitions. Biroli et al. define a dynamical phase transition as a discontinuity of the late time behavior of the order parameter as a function of the quench parameter, also see [24–26]. In this section we complement the discussion with the dynamics of the variance $\text{var}(n_+)$ in Figs. 2 and 3. These results are valuable for the understanding of the entanglement dynamics in Sec. 5.

Based on the qualitative behavior of the variance, we distinguish four different regimes in the dynamical phase diagram, as indicated by the Roman numerals in Fig. 1.

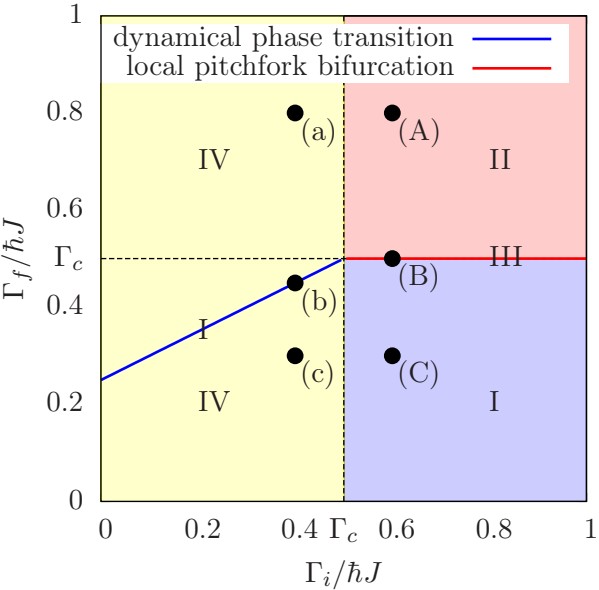

Figure 1: Dynamical phase diagram for the sudden quench $\Gamma_i \to \Gamma_f$ in the Hamiltonian (1) (compare Ref. [39]). Black dots with lower and upper case Latin letters indicate the quenches shown in Figs. 2 and 3, respectively. The different colors and Roman numerals indicate different qualitative behavior of the time evolution of the variance. Region I: exponential growth (cf. Fig. 2 (b) and Fig. 3 (C)); region II: periodic oscillations (cf. Fig. 3 (A)); region III: quadratic growth without squeezing (cf. Fig. 3 (B)); region IV: periodically enhanced squeezing and quadratic growth (cf. Figs. 2 (a) and (c)).

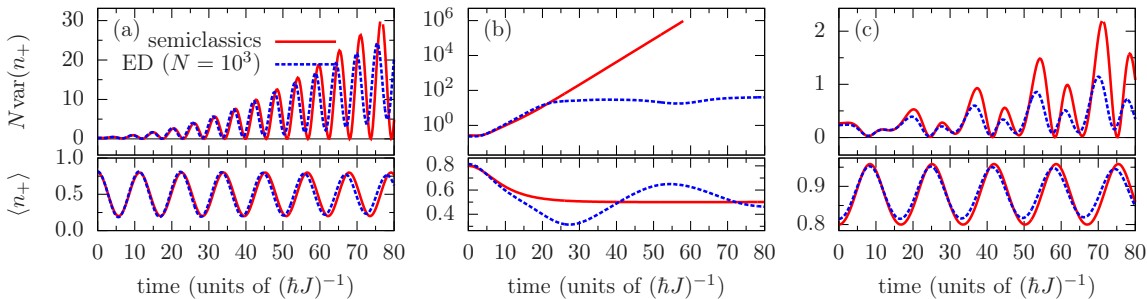

Figure 2: Dynamics of the spin expectation value $\langle n_+ \rangle$ (bottom) and its (rescaled) variance $N \operatorname{var}(n_+)$ (top) after a sudden quantum quench from $\Gamma_i = 0.4$ to $\Gamma_f = 0.8$ (a), $\Gamma_f = 0.45$ (b), and $\Gamma_f = 0.3$ (c) (cf. Fig. 1). The results are obtained by exact diagonalization with $N = 10^3$ (dotted blue line) and by a leading order semiclassical expansion $\lim_{N\to\infty} \langle n_+ \rangle$ and $\lim_{N\to\infty} N \operatorname{var}(n_+)$ (solid red line) according to Eqs. (4) and (5).

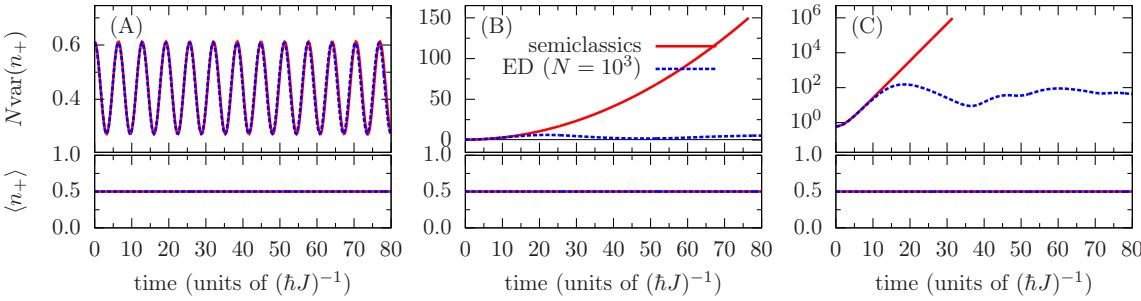

Figure 3: Dynamics of the spin expectation value $\langle n_+ \rangle$ (bottom) and its (rescaled) variance $N \operatorname{var}(n_+)$ (top) after a sudden quantum quench from $\Gamma_i = 0.6$ to $\Gamma_f = 0.8$ (A), $\Gamma_f = 0.5$ (B), and $\Gamma_f = 0.3$ (C) (cf. Fig. 1). The results are obtained by exact diagonalization with $N = 10^3$ (dotted blue line) and by a leading order semiclassical expansion $\lim_{N \to \infty} \langle n_+ \rangle$ and $\lim_{N \to \infty} N \operatorname{var}(n_+)$ (solid red line) according to Eqs. (4) and (5).

## 4.1 Exponential growth regime (I)

For quenches from the paramagnetic phase to the ferromagnetic phase (exemplified by the quench (C) in Fig. 3), as well as for quenches on the critical line of the dynamical phase transition (exemplified by the quench (b) in Fig. 2)), the variance starts to increases exponentially in time before it saturates and shows minor oscillations around a finite value. The saturation process is due to finite size effects and is not captured in the semiclassical result $\lim_{N \to \infty} N \operatorname{var}(n_+)$.

In the case of the quench in Fig. 3 (C) the exponential increase can be readily understood from the fact that the initial wave packet is localized at the hyperbolic critical point $(n_{\text{cl}}, p_{\text{cl}}) = (1/2, 0)$ of the post-quench Hamiltonian, cf. Eq. (11a) [76]. From the point of view of Eq. (8) one can argue as follows. If $\lambda_1 < 0 < \lambda_2$ denote the eigenvalues of the Hessian $H''(\Gamma_f)$ evaluated at the hyperbolic point, then the eigenvalues of $S(t) = \exp(JH''(\Gamma_f)t)$ are $e^{\pm \omega t}$, where $\omega = \sqrt{|\lambda_1 \lambda_2|}$. Hence, the covariance matrix $C(t) = S(t)C(0)S(t)^T$ has an exponentially increasing and an exponentially decreasing eigenvalue in time. For late times, the direction of decreasing variance becomes orthogonal to the stable manifold of the hyperbolic fixed point (a more detailed discussion can be found in Appendix E). For all other directions the exponentially increasing contribution eventually dominates the variance. In particular, $\operatorname{var}(n_+) = C_{11}(t)$ increases exponentially.

For quenches on the critical line (see Fig. 2 (b)), the mean of the initial Wigner distribution lies on a separatrix of the post-quench Hamiltonian. More precisely, the separatrix is a homoclinic orbit and connects the stable and unstable direction of the hyperbolic critical point $(1/2, 0)$ of $H(\Gamma_f)$. As the mean of the Wigner function approaches the hyperbolic fixed point on the separatrix, the dynamics of its variance is dominated by the hyperbolic fixed point and increases exponentially, as discussed above.

## 4.2 Periodic regime (II)

For quenches within the paramagnetic phase (region II in Fig. 1) the post quench Hamiltonian has an elliptic fixed point at $(1/2, 0)$, where the initial Wigner function is localized.

Consequently, the eigenvalues of $S(t)$ are phase factors $e^{\pm i\omega t}$, where $\omega = \sqrt{|\lambda_1 \lambda_2|}$, and the covariance matrix $C(t)$ is $2\pi/\omega$ periodic (see Fig. 3 (A)). Note that the semiclassical result $\lim_{N\to\infty} N \operatorname{var}(n_+)$ agrees with the exact diagonalization data for much later times than in regime (I). Essentially, this is because the Wigner function remains well localized also for late times, which is the key assumption for the validity of the nearby orbit approximation and the rate function expansion.

## 4.3 Quadratic growth regime (III)

The exponential regime (I) and the periodic regime (II) are separated by regime (III) in which the variance increases quadratically. For quenches on this line the initial Wigner function is centered at a degenerate fixed point of the critical post-quench Hamiltonian. The degeneracy leads to the fact that $S(t)$ is a shear matrix whose shear factor scales linearly with time (see Appendix E). Consequently, the eigenvalues of $C(t)$ scale quadratically and inversely quadratic in time. The associated eigenvectors approach the eigenvectors of $H''(\Gamma_f)$ (the eigenvalue zero eigenvector of the Hessian is approached by the quadratically increasing eigendirection of $C(t)$). Along any direction different from the eigendirection in which $C(t)$ decreases, the quadratically increasing contribution dominates for late times, such that the variance increases quadratically in those directions. In particular, $\operatorname{var}(n_+) = C_{11}(t)$ increases quadratically (see Fig. 3 (B)).

Also note that regimes (I), (II) and (III) cannot be distinguished by just looking at the expectation value $\langle n_+ \rangle$. Notwithstanding, its variance behaves qualitatively very different in each case.

## 4.4 Periodically enhanced squeezing regime (IV)

For quenches starting in the ferromagnetic phase and not lying on the critical line of the dynamical phase transition (region IV in Fig. 1), the expectation value oscillates coherently with period $T$. The variance shows quasi-periodic oscillations of the same period $T$ within the envelope of quadratically increasing and inversely quadratic decreasing bounds, Fig. 4. We refer to this behavior as *periodically enhanced squeezing* and *periodically enhanced spreading*. Among all regimes, this is the less intuitive and, to the authors' knowledge, has not been described in the literature so far. In contrast to the regimes (I), (II) and (III) the mechanism is not related to fixed point dynamics of the Hamiltonian flow and therefore genuinely different.

It turns out that the dichotomy of periodically enhanced squeezing and periodically enhanced spreading is the effect of a common cause. As elaborated in Appendix D, the periodicity of the reference orbit allows to apply Floquet's theorem to Eq. (9) and yields $S(t) = P(t)M(t)$ where $P(t)$ is a $T$-periodic two by two matrix and $M(t)$ is a shear matrix with shear factor proportional to time $t$. A non-harmonic Hamiltonian is a necessary condition for the shear factor to be different from zero (see Appendix D). Intuitively, a non-zero shear factor means that two nearby periodic orbits have different periods, which is the rule rather than the exception. An explicit expression of the shear factor is derived in Eqs. (38) and (39). Analogous to regime (III), the eigenvalues of $M(t)C(0)M(t)^T$ scale quadratically and inversely quadratic at late times. Let the corresponding eigenvectors be $|+\rangle(t)$ and $|-\rangle(t)$, respectively. For any fixed initial direction $|v\rangle$, $|w(t)\rangle = P(t)^T |v\rangle$ traverses all directions in the two-dimensional phase space at least once in each period, see Appendix D. As a consequence, $C_v(t) = \langle v|C(t)|v\rangle = \langle w(t)|M(t)C(0)M(t)^T|w(t)\rangle$ has a local minimum and maximum when-

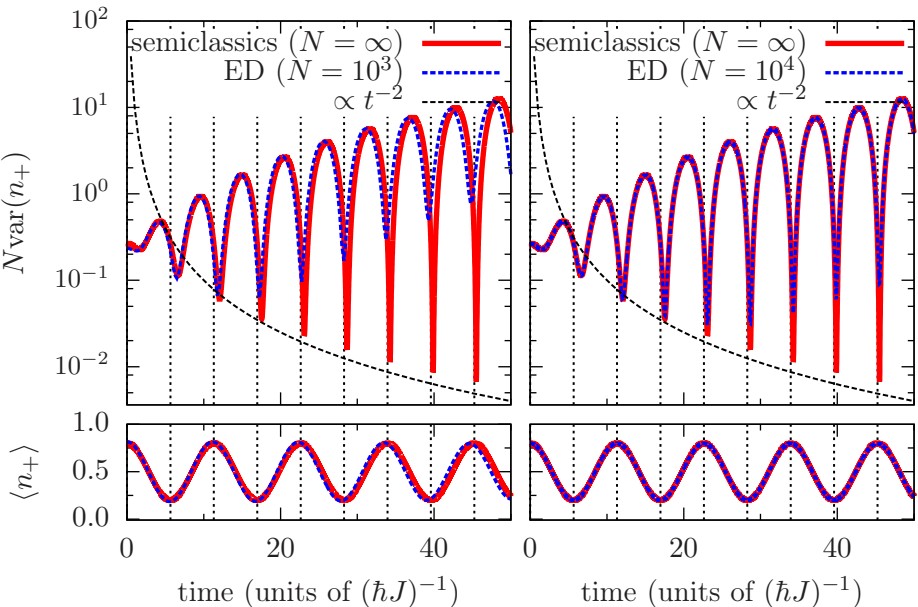

Figure 4: Same quench as in Fig. 2 (a) for $N = 10^3$ (left) and $N = 10^4$ (right). Periodically enhanced squeezing: In the semiclassical limit (solid red line) the minima of var($n_+$) decrease inversely quadratic with time (the dashed black line is a guide to the eye). The positions of the minima approach the turning points of the order parameter (vertical dotted black lines) for late times. The ED data (dashed blue line) agrees with the semiclassical result for early times.

ever $|w(t)\rangle$ aligns with the vector $|-\rangle(t)$ and $|+\rangle(t)$, respectively. This results in the observed periodically enhances squeezing and spreading.

Interestingly, the details of the Hamiltonian do not matter, as long as the reference orbit is periodic and nearby orbits have different periods. In this sense, our observations are universal and to be found in other mean field models, which possess an effective semiclassical two-dimensional phase space description such as the Bose-Hubbard model or the Jaynes-Cummings model on a fully connected lattice [39]. Also, the universality of the periodically enhanced spreading and squeezing shows in the fact that the variance dynamics is qualitatively identical on both sides of the dynamical phase transition, cf. Figs. 2 (a) and (c).

## 4.5 Validity of the mean field approximation in non-equilibrium

We comment on the validity of the semiclassical results. At some point in time, the semi-classical results start to deviate from the exact diagonalization data. A natural question is thus: Up to which timescale can one trust the semiclassical results? This question is really a question about the order of the two limits $N \to \infty$ and $t \to \infty$. If the limit $N \to \infty$ is taken first, the semiclassical results become exact for all times. However, we consider the situation when $N$ is huge but finite, and late times are probed for fixed $N$.

A necessary condition for the validity of the saddle point approximation, on which the semiclassical results (4) and (5) rely, is that $|\psi|^2$ in (3) remains localized on a scale of $1/\sqrt{N}$. More precisely, the leading order saddle point approximation breaks down when the inverse curvature of the rate function at the saddle point is of the order of the saddle point parameter, i.e. $N$.

From the point of view of the nearby orbit approximation, mean field breaks down when the eigenvalues of the covariance matrix $C(t) = S(t)C(0)S(t)^T$ becomes large, such that orbits far away from the reference orbit need to be taken into account. For orbits far away from the reference orbit, the linear approximation of the equations of motion, on which the nearby orbit approximation relies, is inaccurate and errors accumulate. In other words, the nearby orbit approximation breaks down at the (Ehrenfest) timescale $t_E^*$ when the spread of the wave packet reaches the scale $\chi$, on which the Hamiltonian can only be badly approximated to quadratic order. A heuristic estimate of this length scale, motivated by a Moyal bracket expansion, is given by $\chi \sim \sqrt{\partial_x V(x)/\partial_x^3 V(x)}$ [77]. To get the scaling exponent of $t_E^*$ as a function of system size, the order of magnitude of $\chi$ is not crucial. Indeed, for polynomial growth, $\sqrt{\mathrm{var}} \sim t^\alpha/\sqrt{N}$, the condition $\sqrt{\mathrm{var}} \lesssim \chi$ implies $t_E^* \sim N^{1/(2\alpha)}$, and for exponential growth, $\sqrt{\mathrm{var}} \sim e^{\lambda t}/\sqrt{N}$, one gets $t_E^* \sim \log N$.

Concerning the different regimes of Fig. 1, we conclude that the semiclassical dynamics is only valid up to short timescales of order $\log N$ for quenches in regime (I) and to times of order $\sqrt{N}$ in regimes (III) and (IV). In regime (IV), in which the order parameter evolves on a periodic orbit, the effect of anharmonic terms in the Hamiltonian is twofold. First, anharmonic terms in the Hamiltonian inevitably cause the wave packet to spread, and squeeze within quadratically increasing, and inversely quadratic decreasing bounds. Second, as the variance of the wave packet becomes of the order of $\chi$, the anharmonic terms cause the breakdown of the mean field approximation.

We emphasize these findings. Even in fully connected lattice model, for which one believes mean field models to yield reliable results, the out of equilibrium mean field dynamics can already start to break down on a relatively short timescale of order $\sqrt{N}$ (regimes III and IV), and even $\log N$ (regime I).

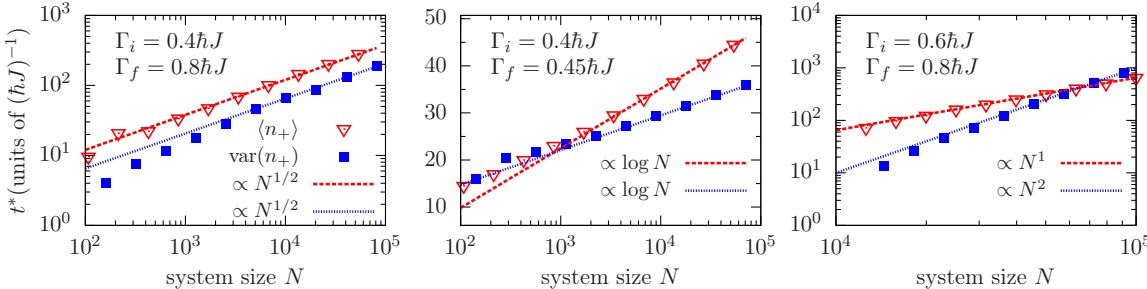

Figure 5: Timescale of validity for the mean field approximation as a function of system size $N$. First instant of time $t^*$, when the leading order correction to the mean field order parameter $\langle n_+ \rangle$ (red open triangles), cf. Eq. (4a), and the mean field variance $\text{var}(n_+)$ (blue filled squares), cf. Eq. (5a), exceeds a fixed, but arbitrarily chosen threshold. The dashed red and dotted blue lines are guides to the eye. Square root scaling, $t^* \propto \sqrt{N}$, for quench (a) in regime IV (left), logarithmic scaling, $t^* \propto \log N$, for quench (b) in regime I (middle), linear scaling, $t^* \propto N$, for quench (A) in regime II (right), cf. Fig. 1. Note, the left and right plots are double-logarithmic, the middle plot is semi-logarithmic.

To confirm this heuristic intuition numerically, we investigate the first time instant $t^*$ at which the leading order correction to the expectation and the variance of the order parameter becomes larger than a arbitrary and fixed threshold, see Fig. 5. These correction terms are functions of $\Re f_3$ and $\Re f_4$, cf. Eq. (32), whose evolution via (31) are sensitive to anharmonic terms of the Hamiltonian.

Quenches within the paramagnetic phase (regime II), where the wave packet is centered at a stable fixed point, are special for two reasons. First, due to spin-flip symmetry $n_+ \mapsto (1 - n_+)$, the expectation value $\langle n_+ \rangle = 1/2$ predicted by mean field is 'accidentally' exact, independent of the system size $N$, and for all times. Second, the evolution of the variance to leading order as given by Eq. (7), depends only on the harmonic part of the Hamiltonian, and is bounded for all times. Despite these facts, one cannot trust the mean field predictions to arbitrarily late times. This becomes apparent, when corrections to the variance are considered, which become significant in size at time $t^* \sim N^2$, see Fig. 5. Spin-flip symmetry implies that all corrections to the mean field limit of $\langle n_+ \rangle$ vanish exactly. To probe the validity of the mean field result for practical purposes, we break the symmetry by adding a term $\epsilon S_z$ to the post-quench Hamiltonian with an infinitesimal longitudinal field $\epsilon$. Then, correction terms to the expectation value build up to a non-negligible contribution on timescales of $t^* \sim N$ being linear in system size, see Fig. 5. The quadratic scaling $t^* \sim N^2$ for the variance corrections is not affected by the symmetry breaking.

## 5  Entanglement dynamics

The ground state entanglement entropy of the fully connected transverse field Ising model has been computed numerically for finite system sizes [78] and analytically in the thermodynamic limit [79] by applying the Holstein-Primakoff [80] transformation and expanding the Hamiltonian in the reciprocal system size. One of the motivations to study the ground state entanglement entropy is its scaling behavior at quantum critical points [81, 82]. A change in

scaling of the mutual information at criticality has also been observed for non-zero temperature thermal density matrices [83].

Entanglement dynamics has been investigated in long-range models with power law interaction, such as harmonic oscillator chains [84], fermionic hopping models [27], spin models [18, 28, 36, 37], and disordered models [19]. Entanglement dynamics as measured by the one-tangle and the concurrence has been investigated in [36] for the fully connected transverse field Ising model.

In the literature so far, the entanglement dynamics has been investigated mainly for fully polarized initial conditions [18, 27, 36]. Since we are ultimately interested in the entanglement entropy of time evolved pre-quench ground states, we follow a different, though related, approach. We will systematically discuss the entanglement dynamics in the dynamical phase diagram of the sudden quench setup. One advantage is that the quantitative connection between entanglement and spin squeezing is apparent in our approach.

## 5.1 Bipartition and reduced density matrix

We want to compute the bipartite entanglement entropy relative to the bipartition $\mathcal{H}_N = \mathcal{H}_{N_A} \otimes \mathcal{H}_{N_B}$. That is, we divide the set of $N = N_A + N_B$ spins into two disjoint sets containing $N_A$ and $N_B$ spins, respectively. Due to the fully connected geometry, the particular choice of the separation into $A$ and $B$ is arbitrary. But once a choice is made, it is fixed over the course of time. Each of the two factors, $\mathcal{H}_{N_A}$ and $\mathcal{H}_{N_B}$, contains a $(N_A + 1)$ and $(N_B + 1)$-dimensional permutation invariant Dicke subspace, respectively. The state $|N_+\rangle$ is expanded in the Dicke basis of the subsystems $A$ and $B$ as

$$|N_+\rangle = \sum_{A_+ + B_+ = N_+} \sqrt{\binom{N_A}{A_+}\binom{N_B}{B_+}\bigg/\binom{N}{N_+}} |A_+\rangle|B_+\rangle. \tag{12}$$

The summation is over all nonnegative integers $0 \le A_+ \le N_A$ and $0 \le B_+ \le N_B$ obeying the constraint $A_+ + B_+ = N_+$. The decomposition is unique. Essentially, the combinatorial factor

$$\sqrt{\binom{N_A}{A_+}\binom{N_B}{B_+}\bigg/\binom{N}{N_+}} \tag{13}$$

reflects the fact that there are more ways to permute $N_+ = A_+ + B_+$ up-spins among $N = N_A + N_B$ spins than to independently permute $A_+$ and $B_+$ up-spins among $N_A$ and $N_B$ spins, respectively.

We want to prove Eq. (12). How does the permutation invariant state $|N_+\rangle$ split into the two permutation invariant subsystems? Equation (12) follows from the identity

$$\binom{N}{N_+} P\left(|\uparrow\rangle^{\otimes N_+} \otimes |\downarrow\rangle^{\otimes N - N_+}\right)$$

$$= \left(|\uparrow\rangle^{\otimes N_+} \otimes |\downarrow\rangle^{\otimes N - N_+} + \text{proper perm.}\right)$$

$$= \sum_{A_+ + B_+ = N_+} \left(|\uparrow\rangle^{\otimes A_+} \otimes |\downarrow\rangle^{\otimes N_A - A_+} + \text{proper perm.}\right)\left(|\uparrow\rangle^{\otimes B_+} \otimes |\downarrow\rangle^{\otimes N_B - B_+} + \text{proper perm.}\right)$$

$$= \sum_{A_+ + B_+ = N_+} \binom{N_A}{A_+} P\left(|\uparrow\rangle^{\otimes A_+} \otimes |\downarrow\rangle^{\otimes N_A - A_+}\right) \binom{N_B}{B_+} P\left(|\uparrow\rangle^{\otimes B_+} \otimes |\downarrow\rangle^{\otimes N_B - B_+}\right)$$

(by proper permutation we mean only those permutations that lead to different spin configurations, e.g. permutations that permute only up-spins are not included). Thus,

$$
\begin{aligned}
|N_+\rangle &= \binom{N}{N_+}^{1/2} P\left(|\uparrow\rangle^{\otimes N_+} \otimes |\downarrow\rangle^{\otimes N-N_+}\right) \\
&= \sum_{A_++B_+=N_+} \sqrt{\binom{N_A}{A_+}\binom{N_B}{B_+} \Big/ \binom{N}{N_+}} |A_+\rangle|B_+\rangle.
\end{aligned}
$$

A generic pure state in $\mathcal{D}_N$ is the superposition $|\Psi\rangle = \sum_{N_+} \psi(N_+)|N_+\rangle$, and the corresponding density matrix is $\rho(N_+; \tilde{N}_+) = \psi(N_+)\psi^*(\tilde{N}_+)$. We can also expand $|\Psi\rangle$ in the Dicke basis of the bipartite system as $|\Psi\rangle = \sum_{A_+,B_+} \psi_{AB}(A_+, B_+)|A_+\rangle|B_+\rangle$, where

$$
\psi_{AB}(A_+, B_+) = \psi(A_+ + B_+)\sqrt{\binom{N_A}{A_+}\binom{N_B}{B_+} \Big/ \binom{N}{A_++B_+}} \tag{14}
$$

follows from Eq. (12). To shorten the notation, we will sometimes write $\psi$ for the coefficient $\psi_{AB}$ of the composite system and distinguish it from the other $\psi$ by the number of arguments. In general, the right hand side of (14) does not factorize into a product of functions depending solely on $A_+$ respectively $B_+$. This shows that the state is entangled. The density matrix associated to $\psi_{AB}$ is $\rho_{AB}(A_+, B_+; \tilde{A}_+, \tilde{B}_+) = \psi(A_+, B_+)\psi^*(\tilde{A}_+, \tilde{B}_+)$ and the reduced density matrix of subsystem $A$ is $\rho_A(A_+, \tilde{A}_+) = \sum_{B_+} \rho_{AB}(A_+, B_+; \tilde{A}_+, B_+)$.

The expectation value of the magnetization per spin in each subsystem agrees with the magnetization per spin of the total system. That is,

$$
\mathrm{Tr}(\rho_A A_+)/N_A = \langle n_+\rangle. \tag{15}
$$

This is an exact result and follows readily from Eq. (14) and the Vandermonde identity,

$$
\begin{aligned}
\langle A_+/N_A\rangle &= \sum_{A_+,B_+} |\psi(A_+, B_+)|^2 A_+/N_A \\
&= \sum_{A_+,B_+} |\psi(A_+ + B_+)|^2 \frac{A_+}{N_A}\binom{N_A}{A_+}\binom{N_B}{B_+}\Big/\binom{N}{A_++B_+} \\
&= \sum_{N^+} |\psi(N^+)|^2 \binom{N}{N^+}^{-1} \sum_{A_+} \binom{N_A-1}{A_+-1}\binom{N_B}{N^+-A_+} \\
&= \sum_{N^+} |\psi(N^+)|^2 \binom{N}{N^+}^{-1}\binom{N-1}{N^+-1} \\
&= \sum_{N^+} |\psi(N^+)|^2 N^+/N = \langle N^+/N\rangle.
\end{aligned}
$$

Eq. (15) is no longer true for higher moments, e.g. in general $\mathrm{Tr}(\rho_A A_+^2)/N_A \neq \langle n_+^2\rangle$, see Eq. (17).

The discussion so far, is valid for generic states in the Dicke subspace. In the remainder of this paragraph we concentrate on states of large deviation form. In particular, we derive the rate function of the reduced density matrix of the pure state (3). Using Eq. (3) in (14)

yields that $\psi_{AB}$ is also of large deviation form $\psi_{AB}(A_+, B_+) \asymp \exp[-N f_{AB}(a_+, b_+)]$ with rate function

$$f_{AB}(a_+, b_+) = f(\alpha a_+ + \beta b_+) + S_{\alpha\beta}(a_+, b_+)/2. \tag{16}$$

Here, and in the sequel, small letters refer to percental quantities, such as the relative subsystem sizes $\alpha = N_A/N$ and $\beta = N_B/N$, and the fraction of up-spins $a_+ = A_+/N_A$ and $b_+ = B_+/N_B$ in subsystem $A$ and $B$, respectively. The multiplicative combinatorial factor (13) translates to the additive entropic contribution $S_{\alpha\beta}$ in (16). It follows readily from Stirling's formula that $S_{\alpha\beta}(a_+, b_+) = H_2(\alpha a_+ + \beta b_+) - \alpha H_2(a_+) - \beta H_2(b_+)$, where $H_2(x) = -x \log x - (1-x) \log(1-x)$ is the classical binary Shannon entropy. Due to the concavity of the Shannon entropy, $S_{\alpha\beta}(a_+, b_+)$ is non-negative and vanishes if and only if $a_+$ and $b_+$ are equal. In other words, fluctuations leading to $a_+ \neq b_+$ are exponentially suppressed. This plays a crucial role in the computation of the reduced density matrix. The term $S_{\alpha\beta}$ has an instructive interpretation. It is the classical information per spin that a demon acquires when splitting $N = N_A + N_B$ spins, containing exactly $N_+ = A_+ + B_+$ up-spins, into two disjoint sets of $N_A$ and $N_B$ spins, each containing $A_+$ and $B_+$ up-spins, respectively. When the demon is blindfolded, the splitting is unbiased and $a_+ = b_+ = n_+$. No information is acquired in this case and $S_{\alpha\beta}(n_+, n_+) = 0$.

Assuming, as before, that $\Re f(n_+)$ has a unique global minimum at $n_{\mathrm{cl}}$, it follows from the properties of $S_{\alpha\beta}$ that $\Re f_{AB}(a_+, b_+)$ has a unique global minimum at $a_+ = b_+ = n_{\mathrm{cl}}$. This is a manifestation of Eq. (15). We expand the composite rate function $f_{AB}$ around this minimum to second order. In this approximation $\psi_{AB}$ is a Gaussian wave function with inverse covariance matrix $N\Gamma^{AB}$,

$$\Gamma^{AB} = f_2 \begin{pmatrix} \alpha^2 & \alpha\beta \\ \alpha\beta & \beta^2 \end{pmatrix} + \frac{1}{2} \frac{1}{n_{\mathrm{cl}}(1 - n_{\mathrm{cl}})} \begin{pmatrix} \alpha\beta & -\alpha\beta \\ -\alpha\beta & \alpha\beta \end{pmatrix}.$$

The latter term is the Hessian matrix of $S_{\alpha\beta}/2$. For future reference, we define $S^* = \alpha\beta/(n_{\mathrm{cl}}(1 - n_{\mathrm{cl}}))$.

To leading order in $1/N$, the kernel of the reduced density matrix $\rho_A$ is again Gaussian and its inverse covariance $\Gamma^A$ is a function of $\Gamma^{AB}$ (see Eq. (48) in Appendix G for details), which yields the variance

$$\mathrm{var}(a_+) = \frac{1}{2N} (\Re f_2)^{-1} + \frac{1}{N} \frac{\beta^2}{S^*} + \mathcal{O}(N^{-2}), \tag{17a}$$

$$\mathrm{var}(p_A) = \frac{\alpha^2}{2N} (\Re(f_2^{-1}))^{-1} + \frac{1}{4N} S^* + \mathcal{O}(N^{-2}), \tag{17b}$$

and covariance $\mathrm{var}(A, B) := \frac{1}{2} \langle AB + BA \rangle - \langle A \rangle \langle B \rangle$,

$$\mathrm{var}(a_+, p_A) = \frac{\alpha}{N} \frac{\Im(f_2)}{\Re(f_2)} + \mathcal{O}(N^{-2}) \tag{17c}$$

of $a_+$ and its conjugate momentum operator $p_A$ by a saddle point approximation. Eqs. (17) should be compared to Eqs. (5). Furthermore, the Wigner function

$$W_A(z) \propto \left[ -\frac{N}{2} z (\Sigma^A)^{-1} z \right] \tag{18}$$

of $\rho_A$ is a Gaussian function of phase space coordinates $z = (a_+, p_A)$, and the two by two covariance matrix $\Sigma^A$ is $N$ independent with $\Sigma_{11}^A/N$, $\Sigma_{22}^A/N$, and $\Sigma_{12}^A/N = \Sigma_{21}^A/N$ given by (17a), (17b), and (17c), respectively. Details are presented in Appendix F.

## 5.2 Entanglement Hamiltonian

Now, we compute the entanglement Hamiltonian $\widehat{H}_E$ w.r.t. the bipartition described above, i.e. we determine the operator $\widehat{H}_E$, such that $\rho_A = \exp(-\widehat{H}_E)$. Note that the Wigner function of $\exp(-\widehat{H}_E)$ is the Gaussian (18). However, we cannot immediately infer that the Wigner function of $\widehat{H}_E$ is the exponent $\frac{1}{2}z(\Sigma^A)^{-1}z$ of $W_A$, because, in general, the Wigner transform and the exponential do not commute (unless the exponent is a linear function in position and momentum). The correct way, to obtain the Wigner function $H_E(z)$ of $\widehat{H}_E$ from $W_A$, is to compute the star-exponential $[\exp^*(-H_E)](z) := \sum_{n=0}(-H_E)^{*n}(z)/n!$ of $H_E$, where $f^{*n}(z)$ denotes the Moyal star product of $n$ factors of $f(z)$, and match the result with $W_A = \exp^*(-H_E)$. For a quadratic function $H_E = \frac{1}{2}zVz$ the star-exponential has been worked out in [85] as $\exp^*(-H_E) \propto \exp\left[-\frac{1}{2}\frac{2}{\sqrt{\det V}}\tanh(\sqrt{\det V}/2)zVz\right]$. We conclude that the entanglement Hamiltonian

$$\widehat{H}_E = \frac{1}{2}zVz + \text{const} \tag{19a}$$

is quadratic, and the two by two matrix

$$V = 2\sqrt{\det \Sigma^A}\,\text{arctanh}\left[(2\sqrt{\det \Sigma^A})^{-1}\right](\Sigma^A)^{-1} \tag{19b}$$

is proportional to the inverse covariance matrix of $\rho_A$. The additive constant results from the multiplicative normalization factor in (18), and can be determined a posteriori by the normalization condition $\text{Tr}\,\rho_A = 1$. It is interesting that in the semiclassical limit the entanglement Hamiltonian of collective spin states takes the simple form of a quantum harmonic oscillator. This is one of the rare cases, when the entanglement Hamiltonian can be computed explicitly.

Next, we compute the entanglement spectrum of $\rho_A$, equivalently, the spectrum of the harmonic oscillator $\widehat{H}_E$. According to Williamson's theorem, there exists a symplectic matrix $S \in \text{Sp}(2)$ such that $S^T V S = \text{diag}(\omega, \omega)$ is diagonal, and $\omega$ is the (unique) symplectic eigenvalue of $V$. Employing this canonical change of coordinates, transforms the entanglement Hamiltonian into the canonical form $\widehat{H}_E = \frac{\omega}{2}\widehat{S}(\widehat{a}_+^2 + \widehat{p}_A^2)\widehat{S}^\dagger$, where $\widehat{S}$ is a metaplectic operator associated to the symplectic matrix $S$. Since the metaplectic operator is unitary, the spectrum is invariant under this transformation, and

$$\text{Spec}(\widehat{H}_E) = \text{const} + \mathbb{N}_0\,\omega. \tag{20}$$

The additive constant combines the zero point energy and the constant in (19a). By Eq. (19b), $\omega$ is related to the symplectic eigenvalue $\lambda$ of $\Sigma^A$ via

$$\omega = 2\,\text{arctanh}[1/(2\lambda)]. \tag{21}$$

Note that $\lambda$ is bounded from below by one half as a consequence of the uncertainty principle, see chapter 13 in [86], so that the argument of the arctanh function is always smaller than or equal to one.

In summary, the entanglement spectrum is equidistant, and a function of the symplectic eigenvalue of the covariance matrix of $\rho_A$. This can be viewed as a refinement of spin squeezing. Spin squeezing subsumes a collection of results around the generic idea that squeezed collective spin states, i.e. states for which the variance of the magnetization in a certain direction is below

the standard quantum limit, are correlated among their elementary spins. These correlations show up in the entanglement of the state w.r.t. a bipartition of the set of elementary spins. Eq. (20) shows that in the large $N$ limit the effect of squeezing, as being measured by the symplectic eigenvalue of the covariance matrix, entails the full entanglement spectrum. To the author's knowledge, this result goes beyond common formulations of spin squeezing. In the following section we discuss the entanglement more closely by investigating the Rényi entanglement entropies.

## 5.3 Rényi entanglement entropies

The $n$th Rényi entanglement entropy $S_A^{(n)} = [\log \text{Tr}(\rho_A^n)]/(1-n)$ follows from the entanglement spectrum (20) and $e^{-\omega} = (2\lambda - 1)/(2\lambda + 1) =: \xi$,

$$
\begin{aligned}
S_A^{(n)} &= \frac{1}{n-1} \log \left[ (\lambda + 1/2)^n - (\lambda - 1/2)^n \right] \\
&= \frac{1}{1-n} \log \frac{(1-\xi)^n}{1-\xi^n}.
\end{aligned}
\tag{22}
$$

In particular, for $n \to 1$, the von Neumann entanglement entropy is

$$
\begin{aligned}
S_A &= (\lambda + \frac{1}{2}) \log(\lambda + \frac{1}{2}) - (\lambda - \frac{1}{2}) \log(\lambda - \frac{1}{2}) \\
&= H_2(\xi)/(1-\xi).
\end{aligned}
\tag{23}
$$

The fact that the von Neumann entropy of a Gaussian density matrix $\rho$ depends only on the symplectic spectrum of the covariance matrix of the Wigner function $W_\rho$, was already noted in [87]. Furthermore, the von Neumann entanglement entropy in fully connected spin models has been obtained by the two-boson formalism in [33]. Our result for the general Rényi entropies in the von Neumann limit is consistent with both of these results. Finally, let us remark that we have computed the Rényi entropies (22) by means of a replica calculation, see Appendix G, yielding the same result and providing yet another consistency check.

More explicitly, $\lambda = \sqrt{\det \Sigma^A}$ follows from Eqs. (17),

$$
\lambda = \sqrt{\frac{1}{4} + \frac{S^*}{4} \left[ \frac{1}{2\Re f_2} + \frac{1}{2\Re \left( f_2^{-1} \right)} \left( \frac{2\alpha\beta}{S^*} \right)^2 - \frac{2\alpha\beta}{S^*} \right]}.
\tag{24}
$$

The fact that this expression contains the variance of $n_+$ and $p$, cf. Eqs. (5), hints to the connection of spin squeezing. This connection is made more explicit below. Remarkably, $\lambda$ and therefore $S_A$ is independent of $N$. This is in contrast to the leading order term of the variance, which decreases as $1/N$. As $N$ increases the wave function becomes more and more concentrated around the classical orbit $n_{\text{cl}}$ in the effective picture, and the expectation value of a permutation invariant observable, such as the mean magnetization per site, is dominated by a single orbit. Quantum fluctuations around the expectation value as measured by the variance decrease and vanish in the limit $N \to \infty$. Nevertheless, the bipartite entanglement entropy, a pure quantum effect, saturates and reaches a non-zero plateau (compare Fig. 6) in the limit $N \to \infty$.

The entanglement entropy is a basis independent quantity that makes only reference to the splitting of the total Hilbert space and is independent of the basis choice in each tensor

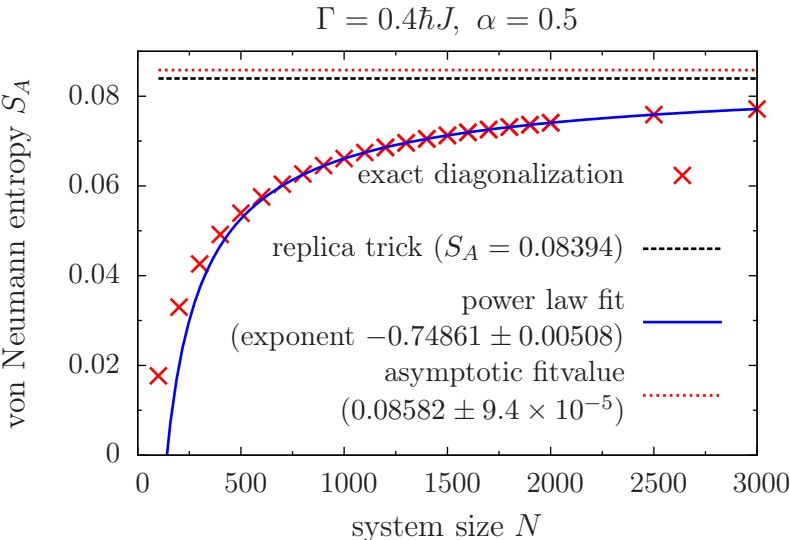

Figure 6: Bipartite entanglement entropy obtained by exact diagonalization as a function of system size (red crosses). In the limit $N \to \infty$ the entanglement entropy saturates to the result given by Eq. (23) (black dashed line), which agrees well with the asymptotic value (red dotted line) of the finite size fit of the exact diagonalization data (blue solid line).

factor. The calculation of $S_A$ in Eq. (23) is done in the eigenbasis of the spin in $z$-direction and leads to the fact that $\lambda$ depends on $n_{\mathrm{cl}}$ and $f_2$, which are not basis independent quantities. As a consequence, the form of Eq. (23) seems to single out a basis. However, this dependence is only an artifact of the representation as we will see below. We seek a more 'covariant' representation of $\lambda$ that is clearly invariant under rotation of the Bloch sphere. It turns out that $\lambda$ is a function of the basis independent spin squeezing parameter $\xi_S^2$ defined below.

One of the first references to establish the connection between entanglement and spin squeezing is the seminal paper of Kitagawa and Ueda [45], also see [46–48]. We review the qualitative argument of Ref. [45] why spin squeezing leads to entanglement. A spin $N/2$ coherent spin state can be viewed a direct product of $N$ identical spin $1/2$ states. Coherent spin states may be considered to be 'most classical states' in the following sense. First, by construction, the individual $1/2$ spins of a coherent spin state are non-entangled among each other. And second, the variance of the magnetization is equally distributed among *all* directions perpendicular to the mean magnetization, such that the uncertainty (i.e. the product of the variance along *any* two orthogonal directions perpendicular to the mean) is minimal. The variance perpendicular to the mean magnetization in a coherent state is referred to as the standard quantum limit (SQL) [88]. Now, a spin state is said to be squeezed if there exists a direction normal to the mean magnetization along which the variance is below the standard quantum limit. In order to lower the variance below the standard quantum limit, correlations among the individual spins need to build up and the individual spins become entangled.

There is a multitude of spin squeezing measures [49]. Among them is what we refer to as the spin squeezing parameter $\xi_S^2$ being the ratio of the minimal to the maximal spin variance

measured along directions perpendicular to the spin expectation value. More specifically, let $\hat{\Omega} = (\cos\phi\sin\theta, \sin\phi\sin\theta, \cos\theta)$ be the direction of the average spin on the Bloch sphere, i.e. $\langle \mathbf{S} \rangle = \hat{\Omega}/2 + \mathcal{O}(1/N)$. We define two directions

$$
\begin{aligned}
\hat{\Omega}_1^\perp &= (-\sin\phi, \cos\phi, 0), \text{ and} \\
\hat{\Omega}_2^\perp &= (-\cos\phi\cos\theta, -\sin\phi\cos\theta, \sin\theta)
\end{aligned}
$$

perpendicular to $\hat{\Omega}$. The covariance matrix of the spin in the subspace spanned by $\hat{\Omega}_1^\perp$ and $\hat{\Omega}_2^\perp$ is given by

$$
C_{ij}^\perp := \langle S_i^\perp S_j^\perp \rangle - \langle S_i^\perp \rangle \langle S_j^\perp \rangle, \tag{25}
$$

where $S_i^\perp = \mathbf{S} \cdot \hat{\Omega}_i^\perp$. The spin squeezing parameter is then defined as

$$
\xi_S^2 = \frac{\min_{\hat{\Omega}^\perp} \langle \hat{\Omega}^\perp | C^\perp | \hat{\Omega}^\perp \rangle}{\sqrt{\det C^\perp}}, \tag{26}
$$

where the minimum is taken over all unit directions $\hat{\Omega}^\perp$ in the plane perpendicular to $\hat{\Omega}$. It turns out (details are given in Appendix H) that $\lambda$ is a function of $\xi_S$ alone, namely $\lambda = \sqrt{1 + \alpha\beta(\xi_S + 1/\xi_S - 2)}/2$, such that the entanglement entropy $S_A$ in Eq. (23) is a function of $\xi_S$ alone.

## 5.4 Dynamics of entanglement

Let us discuss the time dependence of $S_A$ after a quantum quench $\Gamma_i \to \Gamma_f$. How does the entanglement entropy scale in time after a quantum quench in the four different regimes of the dynamical phase diagram in Fig. 1?

We present two related views on the dynamics of entanglement. First, we discuss the intimate connection between the entanglement entropy and the variance of the collective spin state on the Bloch sphere. This point of view establishes the paradigm of spin squeezing. Second, we elaborate on the insight that the entanglement Hamiltonian is a harmonic oscillator whose angular frequency determines the entanglement spectrum and hence all Rényi entanglement measures.

The entanglement dynamics is tightly connected to the dynamics of the variance. We find that $\lim_{N\to\infty} \det NC^\perp = 1/4^2$ (see Appendix H), i.e. there are two directions, call them $\hat{\Omega}_1$ and $\hat{\Omega}_2$, inside the $\hat{\Omega}_1^\perp \hat{\Omega}_2^\perp$ plane such that the uncertainty between $\mathbf{S} \cdot \hat{\Omega}_1$ and $\mathbf{S} \cdot \hat{\Omega}_2$ is minimized to leading order in $N$. Note that the eigenvalues of $C^\perp$ are $\mathrm{var}(\mathbf{S} \cdot \hat{\Omega}_1)$ and $\mathrm{var}(\mathbf{S} \cdot \hat{\Omega}_2)$. If both eigenvalues are exactly equal to the SQL, the variance is equally distributed in the $\hat{\Omega}_1^\perp \hat{\Omega}_2^\perp$ plane and the state is a non-entangled coherent spin state. However, if the variance of the magnetization along, say, $\hat{\Omega}_1$ is larger than the SQL, then the variance in the direction of $\hat{\Omega}_2$ must be below the SQL and the state is squeezed. The variance $\mathrm{var}(n_+)$ of the magnetization in $z$-direction is a lower bound for the maximal eigenvalue of $C^\perp$. Hence, if $\mathrm{var}(n_+)$ increases in time, the minimal eigenvalue of $C^\perp$ must decrease so that the state becomes squeezed and the individual spins become entangled. This is a qualitative reasoning why the von Neumann entanglement entropy increases as $\mathrm{var}(n_+)$ increases.

The quantitative relation between the variance $\mathrm{var}(n_+)$ and the entanglement entropy $S_A$ follows from Eq. (24). As the variance increases, $\xi$ in Eq. (23) approaches one from below and $S_A$ increases. In particular, if the variance $\mathrm{var}(n_+)$ grows exponentially with time as in

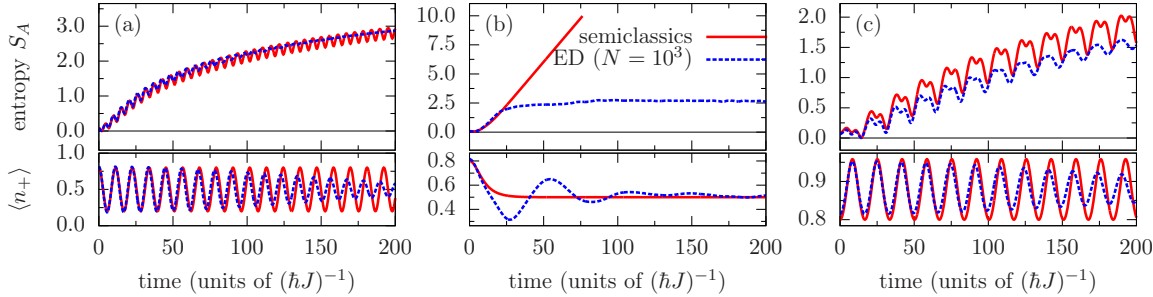

Figure 7: Dynamics of the von Neumann entanglement entropy $S_A$ for a symmetric bipartition (top), and spin expectation value $\langle n_+ \rangle$ (bottom) after a sudden quantum quench from $\Gamma_i = 0.4$ to $\Gamma_f = 0.8$ (a), $\Gamma_f = 0.45$ (b), and $\Gamma_f = 0.3$ (c) (cf. Fig. 1). The results are obtained by exact diagonalization with $N = 10^3$ (dotted blue line) and by a leading order semiclassical expansion $\lim_{N\to\infty}\langle n_+\rangle$ and $\lim_{N\to\infty} S_A$ (solid red line) according to Eqs. (4) and (23).

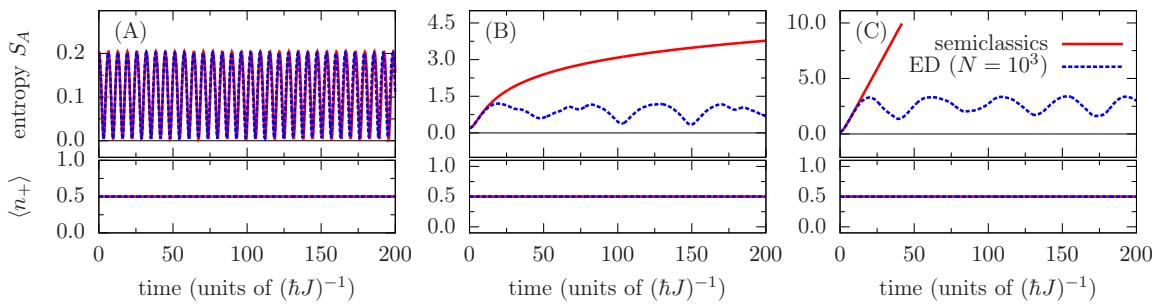

Figure 8: Dynamics of the von Neumann entanglement entropy $S_A$ for a symmetric bipartition (top), and spin expectation value $\langle n_+ \rangle$ (bottom) after a sudden quantum quench from $\Gamma_i = 0.6$ to $\Gamma_f = 0.8$ (A), $\Gamma_f = 0.5$ (B), and $\Gamma_f = 0.3$ (C) (cf. Fig. 1). The results are obtained by exact diagonalization with $N = 10^3$ (dotted blue line) and by a leading order semiclassical expansion $\lim_{N\to\infty}\langle n_+\rangle$ and $\lim_{N\to\infty} S_A$ (solid red line) according to Eqs. (4) and (23).

regime (I), the entanglement entropy increases linearly, cf. Fig. 7 (b) and Fig. 8 (C). In regime (II) where the variance oscillates and remains bounded over time, the entanglement entropy shows bounded oscillations, cf. Fig. 8 (A). Logarithmic entanglement growth can be observed in regimes (III), cf. Fig. 8 (B), and (IV), cf. Fig. 7 (a) and Fig. 7 (c), and is a consequence of the quadratic increase of the variance. Similar results for the von Neumann entropy and a similar semiclassical interpretation were obtained in [33] by the different, though related, approach of the two-boson method.

Another point of view is fascilitated by the fact that the entanglement Hamiltonian (19) is a harmonic oscillator with angular frequency $\omega$. How does $\omega(t)$ change as a function of time after the quench? Figures 9 and 10 display the time dependence of $\omega(t)$ for the quenches of Fig. 1. By inspection of (23) one infers that $\omega$ decreases as the von Neumann entropy $S_A = H_2(e^{-\omega})/(1 - e^{-\omega})$ increases, and similarly for the other Rényi entropies in (22). This

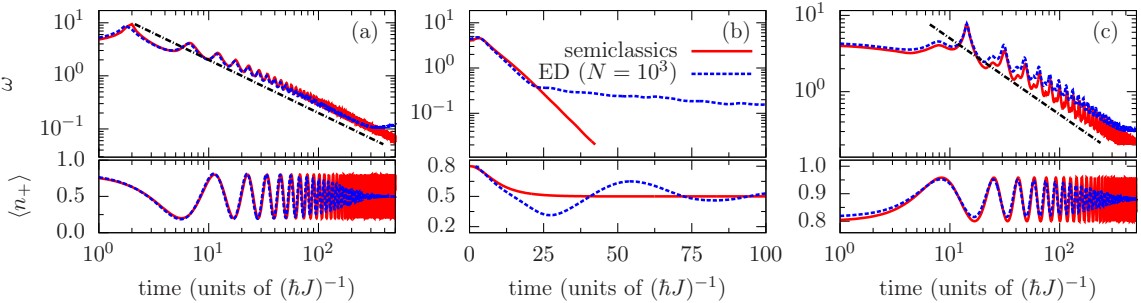

Figure 9: Dynamics of the angular frequency $\omega$ of the entanglement oscillator (19) for a symmetric bipartition (top), and spin expectation value $\langle n_+ \rangle$ (bottom) after a sudden quantum quench from $\Gamma_i = 0.4$ to $\Gamma_f = 0.8$ (a), $\Gamma_f = 0.45$ (b), and $\Gamma_f = 0.3$ (c) (cf. Fig. 1). The results are obtained by exact diagonalization with $N = 10^3$ (dotted blue line) and by a leading order semiclassical expansion $N \to \infty$ according to Eq. (4), and Eqs. (21) and (24). Note that $\omega$ is plotted with double logarithmic scaling in (a) and (c), while the plot of $\omega$ in (b) is a semi-log plot. The black dashed line $\propto 1/t$ in (a) and (c) is a guide to the eye.

has a natural interpretation in the language of thermodynamics. Instead of thinking of the angular frequency as a time dependent quantity, one can equivalently keep it at a fixed value, say $\omega_0 = \omega(0)$, and put the time dependence into a scaling factor $\beta(t)$, that is $\omega(t) = \omega_0 \beta(t)$. We refer to the scaling factor as the inverse entanglement temperature to emphasize the thermodynamic analogy. Small $\omega(t)$ corresponds to large entanglement temperature, so that many entanglement Hamiltonian eigenstates are similarly occupied, and the entanglement entropy of the reduced density is large. On the contrary, large $\omega(t)$ corresponds to small temperature implying that the occupation of high entanglement Hamiltonian eigenstates is suppressed, leading to small entanglement. In the limit $\omega(t) \to \infty$ the entanglement temperature is zero, such that only the ground state is occupied and the reduced density matrix is pure. More quantitatively, the late time asymptotics of $S_A(t) \to \infty$ and $\omega(t) \to 0$ are related by $S_A = -\log(\omega) + 1 + \mathcal{O}(\omega^2)$, cf. Eq. (23). Hence, linear growth $S_A \propto t$ of the entanglement entropy translates to exponential decrease $\omega \propto e^{-ct}$ of the angular frequency, equivalently, to exponential increase of the entanglement temperature, cf. Fig. 9 (b) and Fig. 10 (C). Logarithmic growth $S_A \propto \log(t)$ implies reciprocal decay $\omega \propto 1/t$, equivalently, linear growth of the entanglement temperature, cf. Fig. 9 (a), (c) and Fig. 10 (B).

We close the discussion of the entanglement dynamics by noting a curious implication for local operations and classical communication (LOCC) protocols. A well known theorem in quantum information theory, see e.g. chapter 12 in [89], states that a pure state $\psi_1^{AB}$ on a bipartite Hilbert space $\mathcal{H}_A \otimes \mathcal{H}_B$ can be transformed into another pure state $\psi_2^{AB}$ by means of a LOCC protocol if, and only if, the sequence of eigenvalues of $\mathrm{Tr}_B |\psi_1^{AB}\rangle\langle\psi_1^{AB}|$ is majorized by the sequence of eigenvalues of $\mathrm{Tr}_B |\psi_2^{AB}\rangle\langle\psi_2^{AB}|$. Valid operations of LOCC protocols include measurements and manipulations of the quantum state by operators that act non-trivially only on one of the two factors $\mathcal{H}_A$ and $\mathcal{H}_B$ at a time, and classical processing of the measured information.

We apply this theorem to the collective spin states $\psi_{AB}(t_1)$ and $\psi_{AB}(t_2)$ at two different instants of time $t_1$ and $t_2$ after the quantum quench. In the large $N$ limit, and for times

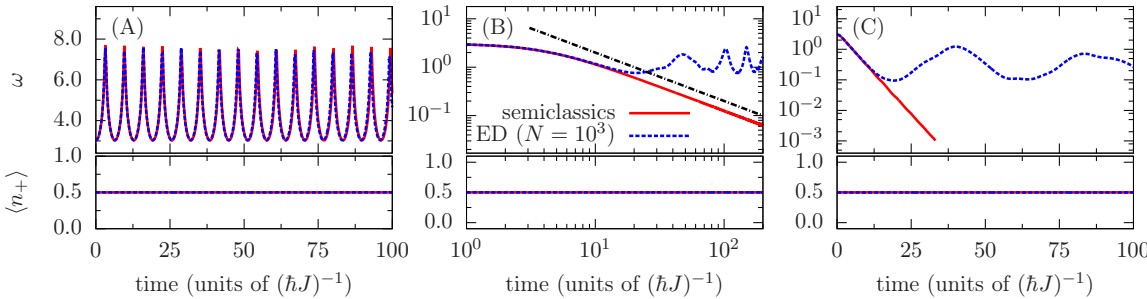

Figure 10: Similar to Fig. 9, showing the dynamics after a quantum quench from $\Gamma_i = 0.6$ to $\Gamma_f = 0.8$ (A), $\Gamma_f = 0.5$ (B), and $\Gamma_f = 0.3$ (C) (cf. Fig. 1). Note, the $\omega$ plot in (B) uses double logarithmic scaling, while the plot of $\omega$ in (C) is a semi-log plot. The black dashed line $\propto 1/t$ in (B) is a guide to the eye.

$t_1, t_2$ when the semiclassical analysis is valid, the non-increasing sequence of eigenvalues of the reduced states $\rho_A(t_1)$ and $\rho_A(t_1)$ is given by

$$P_1 = (1 - \xi_1, \xi_1(1 - \xi_1), \xi_1^2(1 - \xi_1), \cdots), \text{ and}$$
$$P_2 = (1 - \xi_2, \xi_2(1 - \xi_2), \xi_2^2(1 - \xi_2), \cdots),$$

respectively, where $\xi_i = e^{-\omega(t_i)}$, according to Eqs. (20) and (52). $P_1$ is majorized by $P_2$, i.e. $(1 - \xi_1) \sum_{j=0}^{k} \xi_1^j \leq (1 - \xi_2) \sum_{j=0}^{k} \xi_2^j$ for all integers $k \in \mathbb{N}_0$, if, and only if, $\xi_1 \geq \xi_2$, equivalently $\lambda_1 \geq \lambda_2$, where $\lambda_i = (1 + \xi_i)/(1 - \xi_i)/2$ is the symplectic eigenvalue of the covariance of $\rho_A(t_i)$. We conclude that the unitary time evolution between two instants of time after the quantum quench can be realized by a LOCC protocol, if, and only if the symplectic eigenvalue $\lambda$ of the covariance of $\rho_A$ is non-increasing. On the contrary, when $\lambda$ increases between two instants of time, the time evolution cannot be realized by LOCC operations.

# 6   Conclusion

Interesting quantum many body systems, for which relevant quantities can be computed exactly or even approximately, are rare. In this paper, we have examined the fully connected transverse field Ising model as a simple, yet non-trivial, mean field model, which is amenable to a systematic mathematical expansion in inverse system size. This model is not only relevant experimentally, but can also be thought of as a prime model to benchmark the validity of mean field approximations out of equilibrium. Compared to equilibrium, it is less well understood when, i.e. up to which time scales, and how accurate mean field approximations are in non equilibrium situations. In the fully connected Ising model, a typical example of a mean field system, the approximation breaks down at surprisingly early times, scaling with the square root of the system size. This early breakdown happens away from unstable critical points and is explained on the basis of a dephasing effect leading to a linear in time spreading of the wave packet.

Based on the dynamics of the order parameter, i.e. the expected magnetization, and its

variance we have discussed the dynamical phase diagram Fig. 1 for global quenches in the transverse magnetic field. We have seen how the behavior of the variance allows to discriminate different regions in the dynamical phase diagram, which cannot be distinguished by the order parameter alone.

We confirmed the quantitative connection between the variance, i.e. spin squeezing, and various entanglement measures. Remarkably, the entanglement Hamiltonian can be stated explicitly in the large system limit. The entanglement Hamiltonian is a time dependent harmonic oscillator, whose spectrum is known exactly and determines all Rényi entanglement entropies. The spectrum depends on the harmonic oscillator through the angular frequency, which in turn can be related to the determinant of the (co)variance of the Wigner transform of the wave function. Consequently, in the mean field transverse field Ising model, spin squeezing entails the full entanglement spectrum.

The key ingredient for a coherent picture of the mean field dynamics, as summarized by the dynamical phase diagram, is the interplay between the the expectation value *and* the variance of the order parameter. On the one hand, the variance neatly explains both, first, the (early) breakdown of the mean field approximation, as well as, second, the qualitative behavior of the entanglement entropy dynamics. On the other hand, the dynamics of the variance depends on the behavior of (the mean field limit of) the expectation value. This is demonstrated by the hierarchical structure of the ordinary differential equations governing the dynamics of the expectation value and its variance. Two situations, in which the influence of the expectation value on the variance becomes particularly clear, is, first, when the expectation value is close to a (stable or unstable) fixed point, and, second, when the expectation value follows a closed periodic orbit. The latter case leads to the subtle phenomenon of 'periodically enhanced squeezing and spreading' of the time evolving wave packet.

The energy landscape in an effective semiclassical phase space determines the center and variance of the time evolved wave packet and thereby the expectation value and variance of permutation invariant observables, such as the mean magnetization. When the wave packet is at an unstable fixed point, as for quenches from the paramagnetic (PM) phase to the ferromagnetic (FM) phase, or close to a homoclinic orbit connecting to the unstable fixed point, as for quenches on the critical line of the dynamical phase transition, the variance increases exponentially in time. For quenches from the PM phase to the FM phase the wave packet is centered at a stable fixed point, resulting in bounded oscillations of the variance, akin to the dynamics of a centered Gaussian wave function in an harmonic oscillator. This fixed point becomes degenerate for quenches from the PM phase to the quantum critical point separating the PM and FM phase. As a consequence of this degeneracy, the variance increases quadratically. The three distinct situations, (a) stable non-degenerate fixed point, (b) stable degenerate fixed point, and (c) unstable fixed point, have the exact same order parameter evolution, but can easily be distinguished by the variance.

For quenches starting in the FM phase away from the critical line of the dynamical phase transition, the behavior of the variance is *not* dominated by the fixed point structure of the energy landscape. Instead, we have elaborated how periodic orbits and deviations from it lead to squeezing of the wave packet in the presence of anharmonic terms in the Hamiltonian. This subtle dephasing mechanism leads to oscillations of the variance within an envelope of quadratically increasing and inversely quadratic decreasing bounds. We have referred to this observation as 'periodically enhanced squeezing and spreading'.

By comparing to exact diagonalization, we find perfect agreement for early times. However, even for large system sizes a dephasing mechanism leads to a deviation from the mean

field approximation as time proceeds. The breakdown of mean field occurs at the Ehrenfest time, when the spread of the wave function, as measured by its variance, becomes comparable to the length scale on which anharmonic terms of the Hamiltonian cannot be neglected. As a consequence, qualitatively different dynamical behavior of the variance leads to different scaling of the timescale of validity with system size $N$. In particular, close to unstable fixed points, characterized by exponential increase of the variance, mean field results are only valid up to times scaling logarithmically in system size. For quenches in the regime of periodically enhanced squeezing and spreading, mean field breaks down on timescales of square root order in system size. Hence, also away from unstable critical points, mean field ceases to be valid after comparatively short times, even in large systems. The other extreme is a stable non-degenerate fixed points, at which the harmonic approximation of the Hamiltonian is particularly good, such that mean field remains valid up to times scaling linearly in system size.

Subsequently, we have shown that the entanglement Hamiltonian w.r.t. a bipartition of the spins into two disjoint sets is a harmonic oscillator. In analogy to thermodynamics, the angular frequency of this oscillator can be interpreted as the inverse entanglement temperature, which determines the entanglement spectrum, and thereby all Rényi entanglement entropies. Equivalently, the entanglement entropies have also been expressed as functions of a spin squeezing parameter, namely the fraction between the maximal and minimal spin variance in directions perpendicular to the mean magnetization. We thereby confirmed the quantitative relation between spin squeezing and the entanglement spectrum.

The observations about the dynamics of the variance, as contemplated in Fig. 1, translate to qualitative different behavior of the entanglement entropy as a function of time after the quench. More precisely, polynomial and exponential increases of the variance leads to logarithmic and linear growth of the entanglement entropy, respectively, while bounded oscillations imply bounded entanglement. In particular, the asymptotic growth of the entropy is logarithmic for quenches starting in the FM phase, i.e. in the regime of 'periodically enhanced squeezing and spreading'. For quenches from the PM phase to the FM phase, and from the PM phase to the PM phase the entropy shows linear growth and bounded oscillations, respectively, while the entropy grows logarithmically for quenches to the quantum critical point separating the FM and PM phase. Finally, quenches on the critical line of the dynamical phase transition are characterized by linear growth of entanglement entropy. To summarize, the different regimes of variance growth in the dynamical phase diagram translate to qualitatively different regimes of entanglement growth.

We expect that many of the above results hold more generally for quantum models in the semiclassical limit.

# Acknowledgements

Discussions with Mariya Medvedyeva, Aditi Mitra, Giuseppe Mussardo, Salvatore Manmana, and Vincenzo Alba are greatly acknowledged.

**Funding information**    This work was supported through SFB 1073 (project B03) of the Deutsche Forschungsgemeinschaft (DFG).

# A    Derivation of the effective Hamiltonian

We want to solve the Schrödinger equation

$$i\partial_t|\psi\rangle = (-NJS_z^2/2 - N\Gamma S_x)|\psi\rangle$$

in the permutation invariant Dicke subspace. To this end, we expand the wave function in the Dicke states as $|\psi\rangle = \sum_{N_+} \psi(N_+)|N_+\rangle$ and deduce the differential equation for the coefficients $\psi(N_+) = \langle N_+|\psi\rangle$. We obtain

$$
\begin{aligned}
i\partial_t\psi(N_+) =\ & -N\frac{J}{2}\left(\frac{N_+}{N} - \frac{1}{2}\right)^2\psi(N_+) - N\Gamma\left[\frac{1}{2}\frac{N_+ + 1}{N}\sqrt{\frac{(N - N_+ - 1) + 1}{N_+ + 1}}\psi(N_+ + 1)\right.\\
& \left. + \frac{1}{2}\frac{N - N_+ + 1}{N}\sqrt{\frac{N_+}{N - N_+ + 1}}\psi(N_+ - 1)\right]\\
=\ & -N\frac{J}{2}\left(n_+ - \frac{1}{2}\right)^2\psi(N_+) - N\Gamma\left[\frac{1}{2}\sqrt{(1 - n_+)(n_+ + 1/N)}\psi(N_+ + 1)\right.\\
& \left. + \frac{1}{2}\sqrt{n_+(1 - n_+ + 1/N)}\psi(N_+ - 1)\right].
\end{aligned}
$$

The expression becomes more symmetric when expressed in terms of the magnetization per site $s = (N_+/N - 1/2)$. Note that the magnetization per site can take $(N + 1)$ possible equidistantly distributed values in the interval between $-1/2$ and $+1/2$. Hence, by a slight abuse of notation, we write $\psi(s)$ for $\psi(N_+ = N(s + 1/2))$ and get

$$
\frac{i}{N}\partial_t\psi(s) = -\frac{J}{2}s^2\psi(s) - \Gamma\frac{1}{2}\left[\sqrt{\frac{1}{4} - s^2 + \frac{1/2 - s}{N}}\psi(s + 1/N) + \sqrt{\frac{1}{4} - s^2 + \frac{1/2 + s}{N}}\psi(s - 1/N)\right].
\tag{27}
$$

Introducing the shift operators [1] $\triangle_\pm$ by $(\triangle_\pm\psi)(s) = \psi(s \pm 1/N)$ (with the understanding that $\psi(\pm 1/2 \pm 1/N) = 0$), yields

$$
\frac{i}{N}\partial_t\psi(s) = \left[-\frac{J}{2}s^2 - \frac{\Gamma}{2}\sqrt{\frac{1}{4} - s^2}(\triangle_+ + \triangle_-) + \epsilon_1\triangle_+ + \epsilon_2\triangle_-\right]\psi(s),
\tag{28}
$$

where $\epsilon_{1,2}(s) = \sqrt{\frac{1}{4} - s^2 + \frac{1/2 \mp s}{N}} - \sqrt{\frac{1}{4} - s^2}$ are of order $1/N$. No approximation has been made so far and the last expression describes the exact propagation in the Dicke subspace $\mathcal{D}_N$.

We may now approximate Eq. (28) in the limit of large $N$. The approximation is twofold. First, we assume that the $(N + 1)$ dimensional vector $\psi(s)$ can be approximated by a smooth function of $s$. That is, we assume there is a smooth function $\phi$ defined on the continuous interval $[-1/2, 1/2]$ such that $\psi(s) = \phi(s) + \mathcal{O}(1/N)$ for all $s \in \{-1/2, -1/2 + 1/N, \dots, 1/2\}$. Under this assumption we may replace the shift operators $\triangle_\pm$ by the formal expression $e^{\pm\partial_s/N}$.

---

[1] Let the shift operators $\triangle_\pm$ on $\mathbb{C}^{N+1}$ be defined by $\triangle_+\psi = (\psi_1, \dots, \psi_N, 0)$ and $\triangle_-\psi = (0, \psi_0, \dots, \psi_{N-1})$. It is easy to see that $\triangle_+$ and $\triangle_-$ are adjoints of each other. More generally, the adjoint of $\mathrm{diag}(g)\triangle_\pm$ is $\mathrm{diag}(\triangle_\mp\bar{g})\triangle_\mp$, where $g$ is the kernel of the diagonal operator $\mathrm{diag}(g)$. Hence, the operator $(g_1\triangle_+ + g_2\triangle_-)$ is Hermitian if $g_1 = \triangle_+\bar{g}_2$. Now, for $g_{1,2}(s) = \sqrt{\frac{1}{4} - s^2 + \frac{1/2 \mp s}{N}}$ one has $g_1(s) = \bar{g}_2(s + 1/N)$, which confirms that the operator on the right hand side of Eq. (27) is Hermitian.

Second, we only consider the leading terms on the right hand side of Eq. (28), i.e. we drop the $\mathcal{O}(1/N)$ terms. We thus obtain

$$\frac{i}{N}\partial_t \phi(s) = \left[ -\frac{J}{2}s^2 - \Gamma\sqrt{\frac{1}{4} - s^2}\cos(p) \right]\phi(s), \tag{29}$$

where $p = -i\partial_s/N$.

Equation (29) may be interpreted as an effective one dimensional Schrödinger equation for a single fictitious particle governed by the Hamiltonian $H(s,p) = -\frac{J}{2}s^2 - \Gamma\sqrt{\frac{1}{4} - s^2}\cos(p)$. Note that the second term in the Hamiltonian is not Hermitian. This is an artifact of the approximation, in particular of the fact that we have neglect terms of order $1/N$. The total magnetization per site plays the role of the particle's position and the inverse system size, $1/N$, plays the role of an effective Planck constant. In the limit of large $N$, when the effective Planck constant is small, we will therefore apply semiclassical techniques to understand the dynamics imposed by Eq. (29).

# B    Rate function expansion

In this appendix we discuss the dynamics of the rate function in the neighborhood of its minimum and derive Eq. (7). More generally, we derive the differential equations for the Taylor coefficients of the rate function expansion around its minimum. The behavior of the Taylor coefficients determine the leading contribution of the order parameter and its variance, see Eqs. (4) and (5). The main result of this appendix is Eq. (7), which is a simple ordinary differential equation for the curvature of the rate function at the minimum. Remarkably, the curvature does not couple to higher derivatives of the rate function. We derive the more general result that the dynamics of $n$th derivative depends only on derivatives of smaller order than $n$.

The equations of motion for the complex rate function $f(x,t)$ is a nonlinear partial differential equation (PDE)

$$\partial_t f(x,t) = iH(x, i\partial_x f(x,t)), \tag{30}$$

compare Eq. (6). Let us assume that $\Re f(.,t)$ has a unique global minimum $x_{\text{cl}}(t)$ at all times $t$. Instead of solving the full PDE (30), we content ourselves with asking a more humble question: What constraints does the PDE (30) impose on the dynamics of $f$ in the neighborhood of $x_{\text{cl}}$? To answer this question, we expand $f(x,t) = \sum_{n=0} f_n(t)\,[x - x_{\text{cl}}(t)]^n/n!$ in a Taylor series around $x_{\text{cl}}$. Note that the Taylor coefficients

$$f_n(t) = \left.\frac{\partial^n f(x,t)}{\partial x^n}\right|_{x_{\text{cl}}(t)}$$

are time dependent due to two reasons. First, because $f(x,t)$ is explicitly time dependent, and second, because $x_{\text{cl}}(t)$ is in general time dependent. Therefore, the time derivative of $f_n$ gets two contributions,

$$\partial_t f_n(t) = \left.\frac{\partial^n \partial_t f(x,t)}{\partial x^n}\right|_{x_{\text{cl}}(t)} + \left.\frac{\partial^{n+1} f(x,t)}{\partial x^{n+1}}\right|_{x_{\text{cl}}(t)} \dot{x}_{\text{cl}}.$$

Applying Eq. (30) on the first term on the right hand side yields

$$\partial_t f_n = i\partial_x^n H(x, i\partial_x f)\big|_{x_{cl}(t)} + f_{n+1}\,\dot{x}_{cl}. \tag{31}$$

Note that, after evaluating the first term at $x = x_{cl}(t)$, the right hand side is a function of $x_{cl}(t)$ and $\{f_n\}_n$. Therefore, Eq. (31) is a system of coupled first order ordinary differential equations for $\{f_n\}$. Remarkably, as we shall prove below, the right hand side of Eq. (31) only seemingly depends on $f_{n+1}$. Hence, the coupling among the $f_n$ obeys a hierarchical structure in the sense that the equation of motion for $f_n$ only depend on coefficients $f_m$ of lower order $m < n$. As a consequence, the differential equations for the first, say, $n$ coefficients $f_1, \ldots f_n$ close and can be solved exactly.

We now prove the fact that the right hand side of (31) does not depend on $f_m$ with $m > n$ inductively. Starting with $n = 1$, Eq. (31) reads $\dot{f}_1 = iH^{(1,0)}(x_{cl}, if_1) - H^{(0,1)}(x_{cl}, if_1)f_2 + f_2\dot{x}_{cl}(t)$. Here $H^{(n,m)}$ denotes the $n$th and $m$th derivative of $H$ w.r.t. its first and second argument, respectively. By the definition of $x_{cl}(t)$ being the minimum of $\Re f(., t)$, the real part of $f_1$ vanishes identically for all times. Writing $f_1(t) = -ip_{cl}(t)$ for the imaginary part, gives $-i\dot{p}_{cl} = iH^{(1,0)}(x_{cl}, p_{cl}) - H^{(0,1)}(x_{cl}, p_{cl})f_2 + f_2\dot{x}_{cl}(t)$. The real and imaginary part of the last equation are Hamilton's equations of motion

$$
\begin{aligned}
\dot{x}_{cl} &= H^{(0,1)}(x_{cl}, p_{cl}), \\
\dot{p}_{cl} &= -H^{(1,0)}(x_{cl}, p_{cl}),
\end{aligned}
$$

as was already noted in Ref. [39]. Thus, the minimum of the rate function follows the classical trajectory and the rate function expansion is an expansion around the classical limit. Notice that the dependence on $f_2$ is canceled.

Proceeding inductively, it remains to show that the term $f_{n+1}\dot{x}_{cl}$ on the right hand side of (31) is canceled for $n \geqslant 2$. In fact, the only term in the expression $i\partial_x^n H(x, i\partial_x f)\big|_{x_{cl}(t)}$ containing $f_{n+1}$ is $-H^{(0,1)}(z_{cl}, p_{cl})f_{n+1}$. This term cancels the term $f_{n+1}\dot{x}_{cl}(t)$ due to the equations of motion, which concludes the claim.

In particular, using $n = 2$ in Eq. (31), gives the dynamics of $f_2$ in terms of the quadratic form

$$\partial_t f_2 = -i(-i, f_2)H''(-i, f_2),$$

$H''$ being the Hessian matrix of the Hamiltonian evaluated at the classical trajectory $(x_{cl}, p_{cl})$. This result is used to investigate the dynamics of the variance according to Eq. (5). It is also the starting point to prove the equivalence to the classical nearby orbit approximation, see Appendix C.

We close this appendix by stating the next to leading order extension of Eqs. (4a) and (5a),

$$\langle n_+ \rangle = n_{cl} - \frac{g_3}{4g_2^2 N} + \mathcal{O}(N^{-2}), \tag{32a}$$

$$\mathrm{var}(n_+) = \frac{1}{2g_2 N} - \frac{g_4}{8g_2^3 N^2} + \frac{g_3^2}{4g_2^4 N^2} + \mathcal{O}(N^{-3}), \tag{32b}$$

where $g_n$ denotes the real part of $f_n$. These equations follow from a next to leading order saddle point approximation.

## C  Nearby orbit vs. large deviation

The purpose of this appendix is to show that the variance as computed within nearby orbit approximation, cf. Eq. (8), is identical to the result obtained by leading order rate function expansion, cf. Eq. (5). In the sequel, we write $H'' = H''(z_r(t))$ for the Hessian matrix of the classical Hamiltonian $H \colon \mathbb{R}^{2n} \to \mathbb{R}$ evaluated at the reference orbit. The reference orbit is the solution of the equations of motion $\dot{z}_r = JH'(z_r)$ with initial condition $z_r(0) = z_0$. For the sake of simplicity, we restrict to the case $n = 1$. All arguments apply for $n > 1$ as well, but the calculation becomes more lengthy.

More precisely, let $C_{\mathrm{NO}}(t) = S(t)C_{\mathrm{NO}}(0)S(t)^T$ be the nearby orbit covariance matrix, where $S(t)$ is the fundamental matrix of the differential equation $\dot{S}(t) = JH''S(t)$ with $S(0) = \mathrm{id}$. And, let

$$
C_{\mathrm{LD}}(t) = \frac{1}{2N} \begin{pmatrix} (\Re f_2)^{-1} & -\Im f_2/\Re f_2 \\ -\Im f_2/\Re f_2 & 1/\Re(f_2^{-1}) \end{pmatrix}
$$

be the covariance matrix as obtained within the large deviation formalism (see Appendix B), where $\partial_t f_2 = -i(-i, f_2)H''(-i, f_2)$, see Eq. (7). We prove the following claim: If the two covariance matrices initially coincide, that is $C_{\mathrm{NO}}(0) = C_{\mathrm{LD}}(0)$, then they agree for all later times as well, i.e. $C_{\mathrm{NO}}(t) = C_{\mathrm{LD}}(t)$ for all $t$.

We look at the difference $D(t) = C_{LD}(t) - C_{NO}(t)$ between the covariance matrix in large deviation and nearby orbit approximation. By assumption, one has $D(0) = 0$. It remains to show that $D(t) = 0$ for all $t > 0$. The derivative of $C_{\mathrm{NO}}(t)$ is

$$
\frac{d}{dt}C_{\mathrm{NO}}(t) = JH''C_{\mathrm{NO}}(t) - C_{\mathrm{NO}}(t)H''J. \tag{33}
$$

The time derivative of each matrix element of $C_{LD}$ follows from Eq. (7):

$$
\begin{aligned}
\frac{d}{dt}(\Re f_2)^{-1} &= -(\Re f_2)^{-2}\Re \dot{f}_2 \\
&= -(\Re f_2)^{-2}\Re\left[-i(-i, f_2)H''(-i, f_2)^T\right] \\
&= -2(\Re f_2)^{-2}\left[\Re(-i, f_2)H''\Im(-i, f_2)^T\right] \\
&= 2\begin{pmatrix} 0, & 1 \end{pmatrix}H''\begin{pmatrix} (\Re f_2)^{-1} \\ -\Im f_2/\Re f_2 \end{pmatrix},
\end{aligned}
$$

and similarly, one obtains

$$
\begin{aligned}
\frac{d}{dt}\Re(f_2^{-1}) &= -2\begin{pmatrix} 1, & 0 \end{pmatrix}H''\begin{pmatrix} -\Im f_2/\Re f_2 \\ \Re(f_2^{-1}) \end{pmatrix}, \\
-\frac{d}{dt}\frac{\Im f_2}{\Re f_2} &= -\begin{pmatrix} 1, & 0 \end{pmatrix}H''\begin{pmatrix} (\Re f_2)^{-1} \\ -\Im f_2/\Re f_2 \end{pmatrix} \\
&\quad + \begin{pmatrix} 0, & 1 \end{pmatrix}H''\begin{pmatrix} -\Im f_2/\Re f_2 \\ \Re(f_2^{-1}) \end{pmatrix}.
\end{aligned}
$$

The last three equations can be written in a unified matrix form as

$$
\frac{d}{dt}C_{\mathrm{LD}}(t) = JH''C_{\mathrm{LD}}(t) - C_{\mathrm{LD}}(t)H''J. \tag{34}
$$

Subtracting Eqs. (33) and (34), we see that the difference $D(t) = C_{LD}(t) - C_{NO}(t)$ fulfills the first order differential equation

$$\frac{d}{dt}D(t) = JH''D(t) - D(t)H''J,$$

with initial condition $D(0) = 0$, which is uniquely solved by $D(t) = S(t)D(0)S(t)^T = 0$.

# D    Nearby orbit approximation for periodic orbits

We have investigated the dynamics of the order parameter and its variance in mean field models after a quantum quench in Sec. 4. In regime (IV), cf. Fig. 1, when the order parameter oscillates periodically, the short time dynamics of the variance shows quasi-periodic breathing. The envelope of these quasi-periodic oscillations shows two distinct features. First, the local maxima of the variance increase quadratically with time. Second, the local minima of the variance decrease inversely quadratic with time. We refer to the latter property as periodically enhanced squeezing. In this appendix we explain that the two features are the consequence of a common cause. In particular, we demonstrate how the observations follow from shearing effects of the quasi-probability distribution as a consequence of non-quadratic interaction terms in the Hamiltonian. As we will see, the non-quadratic terms are a *sine qua non* ingredient and the precise form of these terms is not important. This not only illustrates the crucial role of the non-quadratic terms, but also indicates the universality of our results independent of the details of the Hamiltonian. The periodicity of the order parameter is crucial for our argument as it enables the application of Floquet's theorem, which plays a key role.

The periodic squeezing is already captured by the leading order of a rate function expansion. As shown in Appendix C the dynamics of the variance to leading order is identical to the Gaussian covariance as obtained in nearby orbit approximation. We may thus use the phase space picture facilitated by the nearby orbit approximation to gain an intuitive understanding.

We consider the time-independent Hamiltonian $H(z)$ and its associated Hamiltonian flow $T_t(z)$ on the $2n$ dimensional phase space, whose coordinates are denoted by $z = (x, p)$. More specifically, $T_t(z_0)$ is the solution of Hamilton's equations of motion, $\dot{z} = JH'(z)$, that passes through $z_0$ at time $t = 0$. In the sequel, the prime denotes differentiation w.r.t. phase space coordinates $z$ and $J$ is the standard symplectic form. Let $z_r(t) = T_t(z_0)$ be a $T$-periodic reference orbit. When approximated to first order around $z_r$, Hamilton's equations impose the differential equation

$$\dot{\delta z} = JH''\big|_{z_r(t)}\delta z \tag{35}$$

on the deviation $\delta z = (z - z_r)$ from the reference orbit. Equation (35) is a first order non-autonomous differential equation with $T$-periodic coefficients. Consequently, the Floquet theorem [90] can be applied. It states that any fundamental matrix $S(t)$ of Eq. (35) decomposes into the product $S(t) = P(t)e^{tB}$. Here, $P(t)$ is a $T$-periodic complex non-singular $2n$ square matrix and $B$ is a constant complex $2n$ square matrix. We refer to $e^{TB}$ as the monodromy matrix and call its eigenvalues the Floquet multipliers. The Floquet multipliers are unique. From now on, we focus on the fundamental system with initial condition $S(0) = P(0) = \mathrm{id}$.

Formally, this can be written as $S(t) = \mathcal{T} \exp(\int_0^t JH''\big|_{z_r(t')} dt')$, where $\mathcal{T}$ denotes time ordering. Note that $S(t)$ is symplectic because it is the linear approximation to the Hamiltonian flow, $S(t) = T_t'(z)\big|_{z=z_0}$. Consequently, also the monodromy matrix is symplectic.

Importantly, because Eq. (35) is obtained by linearizing the equations of motion around $z_r(t)$, the time derivative $\dot{z}_r(t)$ is a solution of (35). Since $z_r$ is $T$-periodic, so is $\dot{z}_r$. Therefore, at least one of the Floquet multipliers is equal to unity. The corresponding eigenspace is spanned by $\dot{z}_r(0)$ and is tangent to the energy hypersurface at $z_0$ in the direction of the reference orbit. This follows readily. As $\dot{z}_r$ solves (35), it can be written as $\dot{z}_r(t) = S(t)\dot{z}_r(0) = P(t)e^{tB}\dot{z}_r(0)$. The periodicity, $\dot{z}_r(t+T) = \dot{z}_r(t)$, then yields $e^{TB}\dot{z}_r(0) = \dot{z}_r(0)$. Moreover, as $e^{TB}$ is symplectic, the roots of its characteristic polynomial come in inverse pairs. Hence, the characteristic polynomial has at least one second root equals unity (we cannot conclude that there is a second Floquet multiplier equals unity because $e^{TB}$ might not be diagonalizable, see below).

From now on, let us consider the case $n = 1$, when the monodromy matrix is two by two and its characteristic polynomial has a two-fold degenerate root equals one. In an appropriate basis this matrix takes the form of a shear matrix

$$e^{TB} = \begin{pmatrix} 1 & \alpha \\ 0 & 1 \end{pmatrix} \tag{36}$$

with shear factor $\alpha$. The fundamental matrix is only periodic for $\alpha = 0$. This case is for example realized by harmonic Hamiltonians (see below). In general, one has to allow for $\alpha \neq 0$, since the monodromy matrix might not be diagonalizable. An orthonormal basis in which the monodromy matrix takes the form (36) is given by the unit vector tangent to the energy hypersurface at $z_0$ and the unit vector in the direction of $H'(z_0)$. We conclude,

$$S(t) = P(t)M(t), \tag{37a}$$

with shear matrix

$$M(t) = \begin{pmatrix} 1 & \alpha t/T \\ 0 & 1 \end{pmatrix}. \tag{37b}$$

We illustrate the consequences of this finding for localized phase space probability distributions. Consider a Gaussian probability distribution $\mu_0(z)$ initially localized at $z_0$ with covariance $C(0)$. The time evolved distribution at a later time $t$ is given by $\mu_t(z) = \mu_0(T_{-t}z)$. For early times and narrow initial covariance, the nearby orbit approximation predicts that $\mu_t$ is close to a Gaussian distribution centered at $T_t(z_0)$ with covariance $C(t) = S(t)C(0)S(t)^T$ [53,54]. It follows from Eq. (37) that the time evolved covariance is obtained by consecutively shearing and periodically modulating the initial covariance. The shear factor $\alpha t/T$ is proportional to time. The variance in the direction of the unit vector $v$ is then determined by the quadratic form $C_v(t) = \langle v|C(t)|v\rangle$. $C_v(t)$ oscillates within the range set by the eigenvalues of $C(t)$. Using the form of $S(t)$ as contemplated in Eq. (37), one reads off that the oscillatory behavior of $C_v(t)$ comes from the periodic modulation by $P(t)$. The envelope of these oscillations is determined by the shear matrix $M(t)$. For the sake of simplicity, let us assume $C(0) = \text{diag}(\lambda_1, \lambda_2)$ is diagonal in the basis in which Eq. (37b) holds. Then the eigenvalues of $M(t)C(0)M(t)^T$ are given by

$$\lambda_{1,2}(t) = \frac{1}{2}\left[\left(\alpha\tfrac{t}{T}\right)^2 \lambda_1 + \text{tr}\right]\left[1 \pm \sqrt{1 - \frac{4\det}{(\alpha\tfrac{t}{T})^2\lambda_1 + \text{tr}}}\right],$$

where $\mathrm{tr} = \lambda_1 + \lambda_2$ and $\det = \lambda_1\lambda_2$. For late times, $t \gg T$, $\lambda_1(t) = (\alpha t/T)^2\lambda_1 + \mathrm{tr} + \mathcal{O}(t^{-2})$ increases quadratically with time whereas $\lambda_2(t) = \det / \left[(\alpha t/T)^2\lambda_1 + \mathrm{tr}\right] + \mathcal{O}(t^{-6})$ decreases inversely quadratic with time. This explains the quadratic increase and the periodically enhances squeezing of the variance.

## D.1   Interpretation of $\alpha$

In the following we derive an explicit expression for the shearing factor $\alpha$ given in Eqs. (38) and (39) below. We will show that a necessary and sufficient condition to observe shearing is that the period of the reference orbit differs from the period of nearby orbits.

The $T$-periodic reference orbit $z_r(t) = T_t(z_0)$ traverses a level set of the Hamiltonian at energy $E_0 = H(z_0)$. Now, consider an initial deviation from the reference orbit in the direction perpendicular to the energy hypersurface, that is $\delta(0) = \epsilon H'(z_0)/\|H'(z_0)\|^2$ for some infinitesimal $\epsilon$. The normalization is chosen such that the energy of this nearby orbit differs from $E_0$ by $\epsilon$, $H(z_0 + \delta(0)) = E_0 + \epsilon + \mathcal{O}(\epsilon^2)$. For small enough $\epsilon$ the orbit starting at $z_0 + \delta(0)$ is also closed, but in general the period is different from the period of the reference orbit. To leading order in $\epsilon$ the period is given by $T(E_0) + \epsilon T'(E_0)$, where $T(E)$ denotes the period of an orbit at energy $E$ close to the reference orbit. An explicit expression of $T'(E_0)$ is given below in Eq. (39). After time $T$ the initial position $z_0 + \delta(0)$ has evolved to $T_T(z_0 + \delta(0)) = z_0 + S(T)\delta(0) + \mathcal{O}(\epsilon^2)$ under the Hamiltonian flow. By the definition of $\delta(0)$ and Eq. (37) one has $S(T)\delta(0) = \epsilon\alpha\dot{z}_r(0)/\|H'(z_0)\|^2 + \delta(0)$. As the difference $dz = T_T(z_0 + \delta(0)) - (z_0 + \delta(0)) = \epsilon\alpha\dot{z}_r(0)/\|H'(z_0)\|^2 + \mathcal{O}(\epsilon^2)$ is infinitesimal but does not vanish unless $\alpha = 0$, the period of the nearby orbit must be different from $T$ if $\alpha \neq 0$. More precisely, comparing to Hamilton's equations, $dz = dt J H'$, one sees that the period of the nearby orbit differs by $dt = \epsilon\alpha/\|H'(z_0)\|^2 + \mathcal{O}(\epsilon^2)$ from the period of the reference orbit. Together with $dt = \epsilon T'(E_0) + \mathcal{O}(\epsilon^2)$ one obtains

$$\alpha = \|H'(z_0)\|^2 T'(E_0). \tag{38}$$

The shearing factor is proportional to the change of the period of nearby orbits at different energies. The derivative is explicitly given by the integral

$$T'(E_0) = -\int_0^T \frac{H'(JH''J + H'')H'}{\|H'\|^4} dt, \tag{39}$$

where $H'$ and $H''$ are evaluated at $z_r(t)$ and the integration is over the full period of the reference orbit. An application of the two dimensional Stokes theorem yields $T'(E_0) = \iint_{\Sigma(E_0)} \mathrm{div}\left[\frac{(JH''J + H'')H'}{\|H'\|^4}\right] dz$, where the integral is over the surface $\Sigma(E_0)$ enclosed by the periodic orbit $z_r(t)$.

To derive Eq. (39), first note that the period of $z_r(t)$ is the integral $T(E_0) = \int \delta(H(z) - E_0)d^2z$. This follows from the equations of motion and $d^2z = dE\, d\sigma_E(z)/\|H'(z)\|$, where $d\sigma_E(z)$ denotes the surface measure on the energy hypersurface $\{z : H(z) = E\}$:

$$T(E_0) = \int \delta(H(z) - E_0)d^2z = \int \frac{d\sigma_{E_0}}{\|H'(z)\|}$$
$$= \int \frac{\|\dot{z}_r(t)\|}{\|H'(z)\|}dt = \int dt$$

(assuming, for the sake of simplicity, that the level set $\{z : H(z) = E_0\}$ consists of a single connected component given by the reference orbit). Straightforward computation then yields

$$
\begin{aligned}
T(E_0 + \epsilon) &= T(E_0) - \epsilon \int \delta'(E - E_0) \frac{d\sigma_E(z)}{\|H'(z)\|} dE + \mathcal{O}(\epsilon^2) \\
&= T(E_0) + \epsilon \int \left( \frac{\partial}{\partial \epsilon}\Big|_{\epsilon=0} \frac{d\sigma_{E_0+\epsilon}(z)}{\|H'(z)\|} \right) + \mathcal{O}(\epsilon^2) \\
&= T(E_0) + \epsilon \int_0^T \left( \frac{\partial}{\partial \epsilon}\Big|_{\epsilon=0} \frac{\|\dot{z}_{r,\epsilon}\|}{\|H'(z_{r,\epsilon})\|} \right) dt \\
&\quad + \mathcal{O}(\epsilon^2),
\end{aligned}
$$

where $z_{r,\epsilon}(t) = z_r(t) + \epsilon H'(z_r(t))/\|H'(z_r(t))\|^2$ is a parametrization of the hypersurface $\{z : H(z) = E_0 + \epsilon\}$. Using the equations of motion $\dot{z}_r = JH'$, in particular, $\|\dot{z}_r\| = \|H'\|$ and $\dot{z}_r \cdot H' = 0$, eventually gives Eq. (39).

## D.2 Example

In the remainder of this appendix we discuss a family of planar Hamiltonians that are amenable to explicit calculations. The example illustrates that non-harmonic terms in the Hamiltonian are necessary in order to have $\alpha \neq 0$. We investigate the class of classical Hamiltonians that depend on the phase space coordinates $z = (x, p) \in \mathbb{R}^2$ only through its Euclidean distance $\|z\| = \sqrt{x^2 + p^2}$. In other words,

$$
H(z) = h(\|z\|^2/2)
$$

for some function $h \colon \mathbb{R} \to \mathbb{R}$. The distance $\|z\|^2$ is an integral of motion of Hamilton's equations $\dot{z} = JH'(z) = h'(\|z\|^2/2)Jz$. The solution that passes through $z_0$ at $t = 0$ is thus

$$
T_t(z_0) = P(t, z_0)z_0, \tag{40}
$$

where $P(t, z_0) = e^{t\omega(z_0)J}$ and $\omega(z_0) := h'(\|z_0\|^2/2)$. Note that $\{P(t, z_0)\}_t$ is a $T = 2\pi/\omega(z_0)$-periodic one-parameter family in the group of orthogonal matrices. The integral curves are thus circles in phase space, which are traversed at a constant angular velocity $\omega(z_0)$. Generically, $\omega$ depends on the initial position. The angular velocity is only independent of the initial condition if $h'$ is constant, i.e. when the Hamiltonian is quadratic. A non-constant angular velocity leads to shearing effects of probability distributions and shall be explained in the following.

Taking the derivative of Eq. (40) w.r.t. $z_0$ yields

$$
S(t) = P(t, z_0) \cdot \left[ 1 + t\Omega(z_0)Jz_0 \otimes z_0 \right], \tag{41}
$$

$(a \otimes b)_{ij} = a_i b_j$ being the dyadic product and $\Omega(z_0) = h''(\|z_0\|^2/2)$. In the harmonic case, when $h'' = 0$, the last term vanishes and $S(t) = P(t, z_0)$ is periodic in time. Moreover, for $\Omega = 0$, $S(t)$ is an orthogonal matrix and $C(t) = S(t)C(0)S(t)^T$ is $2\pi/\omega$-periodic. As a consequence, the variance along any fixed direction (in particular, along the $x$ and $p$ direction) shows periodic breathing.

We will now focus on the less trivial non-harmonic situation and assume $\Omega(z_0) \neq 0$. Without loss of generality and for the sake of clarity, we set $z_0 = (x_0, 0)$ to obtain

$$
S(t) = P(t, z_0) \begin{pmatrix} 1 & 0 \\ -x_0^2 \Omega t & 1 \end{pmatrix}.
$$

This is of the same general form as predicted by Floquet's theorem in Eq. (37). One can read off the shearing factor $\alpha = -x_0^2 T\Omega$, which agrees with Eqs. (38) and (39). The time evolved covariance $C(t) = S(t)C(0)S(t)^T$ is hence obtained by consecutively shearing and rotating the initial covariance. Whereas the rotation $P(t, z_0)$ is periodic in time, the shearing factor $-x_0^2\Omega t$ is proportional to time. Interestingly, the shearing factor depends only through the curvature $\Omega$ on the Hamiltonian but is independent of other details.

## E    Nearby orbit approximation at fixed points

In the previous appendix D we have discussed the dynamics of the covariance matrix within nearby orbit approximation in the case when the reference orbit is periodic. A limiting case occurs when the period of the reference orbit vanishes, i.e. when the reference orbit is a single critical point $z_0$ of the Hamiltonian, that is $H'(z_0) = 0$. Then, $z_0$ is a fixed point of the Hamiltonian flow and the solution of

$$\dot{S} = JKS$$

is $S(t) = \exp(JKt)$, where $K = H''(z_0)$. Note that $S$ obeys an autonomous differential equation and no time ordering is needed for the exponential. Let us restrict to $n = 1$ when $K$ is a symmetric two by two matrix. The real eigenvalues $\lambda_1$ and $\lambda_2$ of $K$ determine the eigenvalues of $JK$ and therefore the dynamics of $S(t)$. This is only true for $n = 1$ and is a manifestation of the fact that every two by two orthogonal matrix is also symplectic. To see this, let $O$ be the orthogonal matrix that diagonalizes $K$, i.e. $OKO^T = \mathrm{diag}(\lambda_1, \lambda_2)$. Then $OJKO^T = J\mathrm{diag}(\lambda_1, \lambda_2)$, where we have used that $O$ is also symplectic, i.e. $OJO^T = J$ (this is no longer true in general for $n > 1$). This shows that the eigenvalues of $OJKO^T$ and thus the eigenvalues of $JK$ only depend on the eigenvalues of $K$. Note that for $n > 1$ the eigenvalues of $JK$ do not solely depend on the eigenvalues of $K$ but also on the direction of the corresponding eigenvectors. For instance, let $n = 2$ and assume $K$ has two positive and two negative eigenvalues. If the two negative eigendirections lie in the $(x_1, p_1)$ plane, then the classical trajectories close to the fixed point are related to ellipses and all eigenvalues of $JK$ are purely imaginary. However, if the two eigendirections of the negative eigenvalues lie in the $(x_1, x_2)$ plane, then the classical orbits close to the fixed point resemble hyperbolas and all eigenvalues of $JK$ are real. An orthogonal transformation rotating $K$ of the latter case into $K$ of the former case cannot be symplectic.

From now on, we assume $n = 1$ and discuss the following cases: (i) $\lambda_1$ and $\lambda_2$ have the same sign, (ii) $\lambda_1$ and $\lambda_2$ have different signs, (iii) exactly one of $\lambda_1$ and $\lambda_2$ vanishes.

In the first case, $z_0$ is a maximum (negative eigenvalues) or a minimum (positive eigenvalues) of $H$ and the fixed point is elliptic, that is the eigenvalues of $JK$, being $\pm i\sqrt{|\lambda_1\lambda_2|} = \pm i\omega$, are purely imaginary. $S(t)$ is $T = 2\pi/\omega$ periodic and is explicitly given by

$$OS(t)O^T = \begin{pmatrix} \cos\omega t & \sqrt{\lambda_2/\lambda_1}\sin\omega t \\ -\sqrt{\lambda_1/\lambda_2}\sin\omega t & \cos\omega t \end{pmatrix}. \tag{42}$$

As a consequence, the covariance matrix $C(t) = S(t)C(0)S(t)^T$ oscillates periodically in time.

In the second case, $z_0$ is a saddle point of $H$ and the fixed point is hyperbolic, that is the eigenvalues of $JK$, being $\pm\sqrt{|\lambda_1\lambda_2|} = \pm\omega$, are real with opposite signs. Analogous to

Eq. (42), one has

$$OS(t)O^T = \begin{pmatrix} \cosh \omega t & \sqrt{|\lambda_2/\lambda_1|}\sinh \omega t \\ \sqrt{|\lambda_1/\lambda_2|}\sinh \omega t & \cosh \omega t \end{pmatrix}.$$

The stable and unstable manifold of the hyperbolic fixed point are Hamiltonian level sets and cross at the fixed point. Let $|v_-\rangle$ and $|v_+\rangle$ be the unstable and stable manifold, respectively, then

$$S(t) = e^{-\omega t}|v_-\rangle\langle w_-| + e^{\omega t}|v_+\rangle\langle w_+|,$$

where $\langle w_i|v_j\rangle = \delta_{i,j}$, and $|v_\pm\rangle$ and $\langle w_\pm|$ are right and left eigenvectors of $S(t)$, respectively. In general, we have to distinguish right and left eigenvectors, because $S(t)$ is not symmetric (unless $|\lambda_1| = |\lambda_2|$). For late times, the covariance matrix $C(t) = S(t)C(0)S(t)^T$ may be approximated by $C(t) = C_{w_+}e^{2\omega t}|v_+\rangle\langle v_+| + \mathcal{O}(e^{\omega t})$, assuming that $C_{w_+} = \langle w_+|C(0)|w_+\rangle$ does not vanish. In other words, for late times one eigendirection of $C(t)$ approaches the direction of the unstable manifold and the corresponding eigenvalue increases exponentially in time. As the phase space volume is preserved under the Hamiltonian flow ($\det S = 1$), there is also a direction in which the covariance decreases exponentially for large times. Due to the symmetry of $C(t)$, this direction is orthogonal to the direction of exponential spreading and becomes orthogonal to $|v_+\rangle$, i.e. parallel to $|w_-\rangle$, for late times. Note that in general, unless $|\lambda_1| = |\lambda_2|$, $|w_-\rangle$ is *not* the direction of the stable manifold.

In the third case, the fixed point is degenerate and one has

$$OS(t)O^T = \begin{pmatrix} 1 & \lambda_2 t \\ 0 & 1 \end{pmatrix}$$

(w.l.o.g. we assume $\lambda_1 = 0$ and $\lambda_2 \neq 0$). In other words, in the basis in which $K$ is diagonal, $S(t)$ has Jordan normal form and is a shear matrix, compare Eq. (37b). Denoting the eigenvectors of $K$ by $|\lambda_1\rangle$ and $|\lambda_2\rangle$, we write $S(t) = \lambda_2 t|\lambda_1\rangle\langle\lambda_2| + |\lambda_1\rangle\langle\lambda_1| + |\lambda_2\rangle\langle\lambda_2|$, such that for late times $C(t) = (\lambda_2 t)^2 C_{\lambda_2}|\lambda_1\rangle\langle\lambda_1| + \mathcal{O}(t)$, where $C_{\lambda_2} = \langle\lambda_2|C(0)|\lambda_2\rangle$. By the same reasoning as above, we conclude that $C(t)$ has a quadratically increasing and inversely quadratic decreasing eigenvalue whose eigenvectors approach $|\lambda_1\rangle$ and $|\lambda_2\rangle$ for late times, respectively.

## F    Wigner function of Gaussian density

In this Appendix we compute the Wigner function $W_\rho$ of a Gaussian density matrix $\rho$ on $L^2(\mathbb{R}^n)$. This is a generalization of Proposition 242 in [86]. The final result is Eq. (45).

Let the kernel of $\rho$ be

$$\rho(x, x') = \sqrt{\frac{\det(X_{11} + X_{12})}{\pi^n}} \, \exp\left(-\frac{1}{2}(x, x')\Gamma(x, x')\right), \tag{43}$$

where the $2n$ by $2n$, symmetric, inverse covariance matrix $\Gamma = X + iY$ has positive definite real part $\Re\Gamma = X > 0$, and

$$X_{11} = X_{22} \text{ symmetric}, \qquad\qquad Y_{11} = -Y_{22} \text{ symmetric}, \tag{44a}$$
$$X_{12} = X_{21} \text{ symmetric}, \qquad\qquad Y_{12} = -Y_{21} \text{ antisymmetric} \tag{44b}$$

($X_{ij}$ denoting $n$ by $n$ blocks of the two by two block matrix $X$, and similarly for $Y$). Eq. (44) is a consequence of Hermiticity of $\rho$, i.e. $\rho(x, x') = \rho(x', x)^*$, and symmetry of $\Gamma$. The factor $[\det(X_{11} + X_{12})/\pi^n]^{1/2}$ normalizes the trace $\operatorname{Tr}\rho = \int \rho(x, x)d^n x$ to unity (positivity of $X$ guarantees positivity of the radicand).

A lengthy, but straightforward calculation of $W_\rho(x, p) = \int d^n\eta\, \rho(x - \frac{\eta}{2}, x + \frac{\eta}{2})e^{ip\eta}$, using the Fourier transform of the Gaussian $\int d^n x [\det(2\pi C)]^{-1/2} \exp\left(-\frac{1}{2}xC^{-1}x\right)e^{-ipx} = \exp\left(-\frac{1}{2}pCp\right)$, yields

$$W_\rho(z) = 2^n (\det X_+ / \det X_-)^{1/2} \exp(-zGz), \tag{45a}$$

where

$$G = \begin{pmatrix} X_+ + Y_- X_-^{-1} Y_+ & Y_- X_-^{-1} \\ X_-^{-1} Y_+ & X_-^{-1} \end{pmatrix}, \tag{45b}$$

$$G^{-1} = \begin{pmatrix} X_+^{-1} & -X_+^{-1} Y_- \\ -Y_+ X_+^{-1} & X_- + Y_+ X_+^{-1} Y_- \end{pmatrix}, \tag{45c}$$

introducing the short hand notation $X_\pm = (X_{11} \pm X_{12})$, and $Y_\pm = (Y_{11} \pm Y_{12})$, such that $X_\pm^T = X_\pm$, and $Y_\pm^T = Y_\mp$, according to (44). The normalization is $\int W_\rho(z)d^{2n}z/(2\pi)^n = 1$. In other words, $W_\rho$ is (proportional to) a Gaussian with covariance matrix $\Sigma = G^{-1}/2$.

In the special case when $\Gamma_{12} = 0$, the kernel $\rho(x, x')$ factorizes and is the a rank one projection (pure state) onto the $L^2$-normalized Gaussian function $\psi(x) = (\pi)^{-n/4}(\det X_{11})^{1/4}\exp(-\frac{1}{2}x(X_{11} + iY_{11})x)$. Then, (45) agrees with Proposition 242 in [86]. Moreover, if $\Gamma_{12} = 0$, $G$ is positive definite, symplectic, and $G = S^T S$, where

$$S = \begin{pmatrix} X_{11}^{1/2} & 0 \\ X_{11}^{-1/2} Y_{11} & X_{11}^{-1/2} \end{pmatrix} \tag{46}$$

is symplectic. That is, the symplectic spectrum of $G$ is unity.

# G   Replica trick

The von Neumann entanglement entropy of Gaussian states was computed by means of the replica trick in [91, 92]. For the sake of completeness, the computation is reviewed in our notation. The final result is given in Eqs. (50) and (51).

The replica trick allows to compute the von Neumann entropy as the derivative

$$S_{\text{vN}}(\rho_A) = -\partial_n\big|_{n=1} \operatorname{Tr}(\rho_A^n).$$

A variant of this formula,

$$S_{\text{vN}}(\rho_A) = (-\partial_n\big|_{n=1} + 1)\log \operatorname{Tr}(\rho_A^n), \tag{47}$$

has the advantage that $\rho_A$ in Eq. (47) does not need to be normalized. The idea is to find an easy explicit symbolic expression of $\operatorname{Tr}\rho_A^n$ in $n$ and then differentiate this expression w.r.t. $n$. Once $\operatorname{Tr}(\rho_A^n)$ is computed for $n = 1, 2, 3, \ldots$, one also knows all the other Rényi entropies $S_n = \log[\operatorname{Tr}(\rho_A^n)]/(1 - n)$.

Let $\psi$ be a Gaussian wave function on the bipartite Hilbert space $L^2(\mathbb{R}) \otimes L^2(\mathbb{R})$ (the derivation can be generalized to $L^2(\mathbb{R}^d) \otimes L^2(\mathbb{R}^d)$),

$$\psi(x_A, x_B) \sim \exp\left[ -\frac{1}{2} \begin{pmatrix} x_A & x_B \end{pmatrix} \Gamma^{AB} \begin{pmatrix} x_A \\ x_B \end{pmatrix} \right],$$

with complex valued, symmetric, two by two inverse covariance $\Gamma^{AB}$. The reduced density matrix

$$\rho_A(x, y) \sim \exp\left[ -\frac{1}{2} \begin{pmatrix} x & y \end{pmatrix} \Gamma^A \begin{pmatrix} x \\ y \end{pmatrix} \right]$$

is again Gaussian with inverse covariance [11]

$$\Gamma_{11}^A = \Gamma_{11}^{AB} - \frac{1}{2}\Gamma_{12}^{AB}(\Re\Gamma_{22}^{AB})^{-1}\Gamma_{12}^{AB}, \tag{48a}$$

$$\Gamma_{12}^A = -\frac{1}{2}\Gamma_{12}^{AB}(\Re\Gamma_{22}^{AB})^{-1}\overline{\Gamma_{12}^{AB}}, \tag{48b}$$

and, due to the hermiticity of $\rho_A$, $\Gamma_{2,1}^A = \overline{\Gamma_{1,2}^A}$, and $\Gamma_{2,2}^A = \overline{\Gamma_{1,1}^A}$. The trace of $\rho_A^n$ is then proportional to the integral over the $n$ dimensional Gaussian

$$\mathrm{Tr}(\rho_A^n) \sim \int d^n x \exp\left[ -\frac{1}{2} x M x \right] \sim \det(M)^{-1/2} \tag{49}$$

with $M$ being the circulant $n$ by $n$ matrix

$$M = \begin{pmatrix}
\Gamma_{11}^A + \Gamma_{22}^A & \Gamma_{12}^A & 0 & \cdots & 0 & \Gamma_{21}^A \\
\Gamma_{21}^A & \ddots & \ddots & \ddots & & 0 \\
0 & \ddots & & & \ddots & \vdots \\
\vdots & \ddots & & & \ddots & 0 \\
0 & & \ddots & \ddots & \ddots & \Gamma_{12}^A \\
\Gamma_{12}^A & 0 & \cdots & 0 & \Gamma_{21}^A & \Gamma_{11}^A + \Gamma_{22}^A
\end{pmatrix}.$$

This matrix is not symmetric, but as it is contracted with a symmetric tensor in the expression $M_{ij}x_i x_j$, we may replace $M$ by its symmetric part $\tilde{M} = (M + M^T)/2$,

$$\tilde{M} = \begin{pmatrix}
2\Re\Gamma_{11}^A & \Re\Gamma_{12}^A & 0 & \cdots & 0 & \Re\Gamma_{12}^A \\
\Re\Gamma_{12}^A & \ddots & \ddots & \ddots & & 0 \\
0 & \ddots & & & \ddots & \vdots \\
\vdots & \ddots & & & \ddots & 0 \\
0 & & \ddots & \ddots & \ddots & \Re\Gamma_{12}^A \\
\Re\Gamma_{12}^A & 0 & \cdots & 0 & \Re\Gamma_{12}^A & 2\Re\Gamma_{11}^A
\end{pmatrix}$$

(where we have used $\Gamma_{2,1}^A = \overline{\Gamma_{1,2}^A}$, $\Gamma_{2,2}^A = \overline{\Gamma_{1,1}^A}$). The integral in (49) is thus proportional to $\det(\tilde{M})^{-1/2}$. The determinant is known to be [93]

$$
\begin{aligned}
\det(\tilde{M}) &= \prod_{j=0}^{n-1} \left[ 2\Re\Gamma_{11}^A + 2\Re\Gamma_{12}^A \cos(2\pi j/n) \right] \\
&= (2\Re\Gamma_{11}^A)^n \prod_{j=0}^{n-1} \left[ 1 + \Re\Gamma_{12}^A/\Re\Gamma_{11}^A \cos(2\pi j/n) \right].
\end{aligned}
$$

We only need to compute $\det(\tilde{M})$ modulo factors of $n$-th power. This is because $\rho_A$ in Eq. (47) does not need to be normalized and rescaling of $\rho_A$ leads to factors of $n$-th power in $\mathrm{Tr}(\rho_A^n)$ and hence in $\det(\tilde{M})$. Thus, we may drop all global factors of $n$-th power in $\det \tilde{M}$, which we indicate by writing $\sim$ instead of the equality sign. Now, we define $\xi$ by $\Re\Gamma_{12}^A/\Re\Gamma_{11}^A = -2\xi/(1+\xi^2)$ and use the identity $\prod_{j=0}^{n-1}\left[1 + \xi^2 - 2\xi\cos(2\pi j/n)\right] = (1-\xi^n)^2$ to obtain

$$\det(\tilde{M}) \sim (1 - \xi^n)^2.$$

The von Neumann entropy follows from Eq. (47)

$$S_{\mathrm{vN}} = -\log(1 - \xi) - \frac{\xi}{1-\xi}\log(\xi). \tag{50}$$

It is not obvious, but $0 < \xi < 1$ (to be more precise, only $\xi_-$ of the two solutions $\xi_\pm = -\Re\Gamma_{11}^A/\Re\Gamma_{12}^A \pm \sqrt{(\Re\Gamma_{11}^A/\Re\Gamma_{12}^A)^2 - 1}$ obeys this constraint), so that the above expression is always real and positive. The other Rényi entropies are given by

$$S_n = \frac{1}{1-n}\log\frac{(1-\xi)^n}{1-\xi^n}. \tag{51}$$

Eqs. (50) and (51) are equivalent to Eqs. (23) and (22), respectively, upon the identification $\xi = \exp(-\omega) = (2\lambda - 1)/(2\lambda + 1)$.

As a corollary of the result (51), we obtain the spectrum

$$\mathrm{Spec}(\rho_A) = \{(1 - \xi)\xi^j : j \in \mathbb{N}_0\} \tag{52}$$

of the reduced density matrix $\rho_A$. This equation follows from comparing (51) with $\frac{1}{1-n}\log\sum_j \lambda_j^n$, where $\lambda_j$ denotes the sequence of eigenvalues of $\rho_A$. The equations

$$\sum_{j=0}^{\infty}\lambda_j^n = \frac{(1-\xi)^n}{1-\xi^n},$$

for all positive integers $n$, are solved by $\lambda_j = (1 - \xi)\xi^j$. Eq. (52) is consistent with (20).

## H   Spin squeezing and entanglement

In this appendix we compute $\lambda$ (cf. Eq. (24)) as a function $\xi_S$. In the sequel, we write $g_2$ and $-\theta_2$ for the real and imaginary part of $f_2 = g_2 - i\theta_2$. The covariance matrix $C^\perp$ (cf. Eq. (25)) is Hermitian and its real part is

$$\Re C^\perp = \frac{1}{2g_2 N}\begin{pmatrix} \sin^{-2}\theta & \frac{\theta_2}{2} \\ \frac{\theta_2}{2} & \frac{\theta_2^2 + g_2^2}{4}\sin^2\theta \end{pmatrix} + \mathcal{O}(1/N^2). \tag{53}$$

The leading order of the determinant of $C^\perp$ is $(4N)^{-2}$ (independent of $g_2$ and $\theta_2$). This means that the uncertainty between the magnetization in the two directions of the eigenvectors of $C^\perp$ is minimized to leading order,

$$\det C^\perp \geq \frac{1}{4N^2}|\langle \mathbf{S}\cdot\hat{\Omega}\rangle|^2 = \frac{1}{(4N)^2} + \mathcal{O}(1/N^3).$$

In the special case, when the eigenvalues of $C^\perp$ are identical, the uncertainty between the magnetization in any two directions in the $\hat{\Omega}_1^\perp \hat{\Omega}_2^\perp$ plane is minimized. This is the situation of coherent states which are non-entangled (see below).

The determinant and the trace of $C^\perp$ are invariant under rotations of the Bloch sphere, i.e. changes of the quantization axis. Determinant and trace are the only two independent basis independent properties of a two by two matrix. As the leading order of the determinant is constant, the entanglement entropy can only depend on the trace. In fact, Eq. (24) can be rewritten as

$$\lambda = \sqrt{\frac{1}{4} + \alpha\beta\left(N\operatorname{Tr}C^\perp - \frac{1}{2}\right)}. \tag{54}$$

Since $\det C^\perp = (4N)^{-2}$, the trace of $C^\perp$ is bounded from below by $(2N)^{-1}$. More precisely, $\operatorname{Tr}C^\perp = (2N)^{-1}$ if and only if both eigenvalues of $C^\perp$ are identical to $(4N)^{-1}$ (coherent states). The Isotropic variance of $(4N)^{-1}$ at minimal uncertainty is called the standard quantum limit (SQL) [45,49]. In this case $\lambda = 1/2$ and all Rényi entropies vanish (cf. Eq. (22)). This is also consistent with the observation that the symplectic spectrum of the covariance of a Gaussian pure state is one half, see the discussion around Eq. (46).

Let $\lambda_1$ and $\lambda_2$ be the eigenvalues of $C^\perp$ with $\lambda_1 \le \lambda_2$ and $\lambda_1\lambda_2 = (4N)^{-2}$, then (cf. Eq. (26))

$$\xi_S^2 = \sqrt{\lambda_1/\lambda_2} = 4N\lambda_1 = 2N\operatorname{Tr}C^\perp - \sqrt{(2N\operatorname{Tr}C^\perp)^2 - 1},$$

which, together with Eq. (54), gives $\lambda$ as a function of $\xi_S^2$. The von Neumann entanglement entropy (and any other Rényi entanglement entropy) is thus an explicit function of the squeezing parameter $\xi_S^2$.

**Details on the computation of $C^\perp$:** Equation (53) follows from the lengthy calculation of

$$\begin{aligned}
\langle S_x, S_x\rangle_c &= +(\cos\phi\cot\theta)^2\frac{1}{2g_2N} + \cos\theta\cos\phi\sin\phi\frac{\theta_2}{2g_2N} \\
&\quad + \left(\frac{1}{2}\sin\phi\sin\theta\right)^2\frac{\theta_2^2 + g_2^2}{2g_2N} + \mathcal{O}(1/N^2), \tag{55} \\
\langle S_y, S_y\rangle_c &= +(\sin\phi\cot\theta)^2\frac{1}{2g_2N} - \cos\theta\cos\phi\sin\phi\frac{\theta_2}{2g_2N} \\
&\quad + \left(\frac{1}{2}\cos\phi\sin\theta\right)^2\frac{\theta_2^2 + g_2^2}{2g_2N} + \mathcal{O}(1/N^2), \tag{56} \\
\langle S_z, S_z\rangle_c &= +\frac{1}{2g_2N} + \mathcal{O}(1/N^2), \tag{57} \\
\Re\langle S_x, S_y\rangle_c &= +(\cot\theta)^2\sin\phi\cos\phi\frac{1}{2g_2N} + \frac{1}{2}\cos\theta\left(\sin^2\phi - \cos^2\phi\right)\frac{\theta_2}{2g_2N} \\
&\quad - \left(\frac{1}{2}\sin\theta\right)^2\sin\phi\cos\phi\frac{\theta_2^2 + g_2^2}{2g_2N} + \mathcal{O}(1/N^2), \tag{58} \\
\Re\langle S_x, S_z\rangle_c &= -\cos\phi\cot\theta\frac{1}{2g_2N} - \frac{1}{2}\sin\phi\sin\theta\frac{\theta_2}{2g_2N} + \mathcal{O}(1/N^2), \tag{59} \\
\Re\langle S_y, S_z\rangle_c &= -\sin\phi\cot\theta\frac{1}{2g_2N} + \frac{1}{2}\cos\phi\sin\theta\frac{\theta_2}{2g_2N} + \mathcal{O}(1/N^2). \tag{60}
\end{aligned}$$

The results (55) to (60) can be obtained by carefully approximating the expectation values in the state $\psi \asymp e^{-Nf(s)}$ to next to leading order in a saddle point approximation.

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
