# Peer review of "Out of equilibrium mean field dynamics in the transverse field Ising model"

_SciPost Physics_

## Round 2 · Referee Report · Anonymous (Referee 1) · 2019-12-2

Report

In “Out of equilibrium mean field dynamics in the transverse field Ising model”, the authors report on a study on the validity of mean-field approximations in infinite-range interacting quantum spin models out of equilibrium. They focus on the behaviour of quantum fluctuations around the classical orbits describing the mean-field motion, and on the evolution of entanglement.

The manuscript is written in a clear and orderly manner. The motivation for this study are properly explained by the authors. Analytical and numerical results are convincing and presented appropriately.

My assessment is that this manuscript deserves publication as it represents a useful addition to the literature in terms of technical results. However, I find that the authors partly misrepresent previous knowledge in the field, to the point that the manuscript should not be published in the present form. I describe below the two major points of concern. In view of these critical issues regarding the originality of the results, I find that this manuscript does not meet the acceptance criteria of SciPost Physics, but could meet those of SciPost Physics Core.

I) An important part of the results detailed in this work may be understood in terms of basic semiclassical mechanics, i.e., in terms of the Liouville evolution of the phase-space representation of Gaussian quantum fluctuations around the classical orbit. It is a well-known fact that the nonequilibrium dynamics of mean-field quantum models maps onto the semiclassical evolution of a finite number of collective variables, with an effective Planck constant inversely proportional to the system size - this has been studied systematically by B. Sciolla and G. Biroli in Ref. [39] of the manuscript, although it was known much before for spin models. The authors write in the Introduction that “it is not generally known when and how well mean field works.”
and repeat similar statements several times in the text. I find that this statement is misrepresentative: Once the mapping to a system of few semiclassical degrees of freedom is established with $\hbar_{\rm eff} \sim \hbar/N$, as done e.g. in the mentioned article by Sciolla-Biroli, then the validity of the mean-field approximation coincides with the validity of the classical description of a quantum system. It is well-known that the failure of classical description occurs at the Ehrenfest time scale dictated by the spreading of quantum wavepackets, which is in turn dictated by the divergence of classical orbits. 
I emphasize that this is basic, textbook knowledge of the classical limit of quantum mechanics. The manuscript’s contribution here is to report calculations and implications for a paradigmatic infinite-range quantum spin system. For this reason, I think that the statement that the validity of mean-field approximation for the dynamics of fully-connected quantum systems was not previously known, is not correct. Similarly, the statement that the timescale of validity of mean-field is “surprisingly short” is misrepresentative, as this time scale is nothing but the well-known Ehrenfest time scale. Concerning the dynamics of fully-connected quantum models, this is also well-understood: for example, I find that the mentioned article by Sciolla-Biroli contains indications on the time scale of validity of the mean-field description, including the distinction between $\sqrt{N}$ and $\log N$ for integrable and chaotic classical dynamics, respectively. Also, I find that the “Periodically enhanced squeezing regime” described in sec. 4.4, contrary to what the authors write, has been described before (for example by A. Lerose and S. Pappalardi in Ref. [33] of the manuscript in a similar context to that of the present paper).

II) As regards the part on entanglement dynamics, I think that similar concerns can be raised. The fact that the entanglement Hamiltonian turns out to equal to that of a harmonic oscillator with a time-dependent frequency is not unexpected as the authors write in the text, because within the Gaussian approximation (asymptotically exact in the semiclassical limit $N\to\infty$) the reduced density matrix is a Gaussian state of a single boson, i.e., necessarily a thermal state of a quantum harmonic oscillator. In addition, I find that the authors systematically omit in this part the citation of Ref. [33], which contains very similar results. As far as I can see, this occurs several times in the text, and is beyond the boundary of scientific fairness in view of the important overlap with the results of this article. I mention the most important such omissions: 
1) in the Introduction, page 3, when discussing the logarithmic growth in time of entanglement entropy, the authors do not mention at all Ref. [33] which first analytically derived and explained this phenomenon.
2) at the beginning of sec. 5 on entanglement dynamics, when reviewing previous work on entanglement dynamics in long-range interacting systems, citation of Ref. [33] is completely absent.
3) at the end of sec. 5.3, the authors mention the analytical relation between entanglement entropy and spin squeezing but do not cite Ref. [33] where this relation was first derived. 4) the same in the conclusions.

In conclusion, I think that the authors should make an effort to revise the text with the aim of making their claims compatible with previous knowledge in the field, specifically by decreasing the strength of their claims of novelty and unexpectedness, and by appropriately citing the relevant literature in the course of their discussion (including, but preferably not limited to, what I mentioned above).

  • validity: -
  • significance: -
  • originality: -
  • clarity: -
  • formatting: -
  • grammar: -

Author:  Ingo Homrighausen  on 2019-12-16  [id 685]

(in reply to Report 1 on 2019-12-02)
Category:
reply to objection

We thank the referee for his/her careful reading of our manuscript and use this opportunity to address the concerns and criticisms in the same order as they appear in the report.

1) The referee considers our statement “it is not generally known when and how well mean field works” [in non-equilibrium conditions] in the introduction (and similar statements throughout the text) misrepresentative. We agree that in the context of semiclassical models the Ehrenfest time scale dictates when the semiclassical Liouville evolution only poorly describes the full quantum dynamics, and that this is well known. In fact, the Ehrenfest time scale is explicitly mentioned in Sec. 4.5, and in the Conclusion, Sec. 6 of our manuscript. However, mean field approximations are used in many situations where there is no semiclassical limit (or where one is very far away from it), e.g. for S=1/2 spin models in finite or correlated electron models in finite and infinite dimensions. A lot is known about the accuracy of the mean field approximations for equilibrium properties of such models (e.g. based on comparison with numerical results), but much less is known about non-equilibrium. Our statement refers to this observation.

We certainly want to avoid confusion and possible misreading of our statement. We will clarify that the validity of mean-field in non-equilibrium is determined by the Ehrenfest time whenever the mapping to a model with a small effective Planck constant is possible, by adding an earlier reference to the Ehrenfest time. Following this we will mention that mean field approximations are also well established in many other equilibrium situations when there is no such mapping, but that much less is known about their validity in non-equilibrium.

The referee also considers our statement “surprisingly short” times in the context of the validity of mean field for the fully connected model misrepresentative. We will change this to “Ehrenfest times” in order to avoid creating a wrong impression. We agree with the referee that Sciolla & Biroli (Ref. [39]) contains “indications on the time scale of validity of the mean-field description, including the distinction between N^(1/2) and log N […]”, but Ref. [39] does not contain a comprehensive analysis of the Ehrenfest time scale via the variance for different quenches in the dynamical phase diagram as our manuscript does. As regards the “periodically enhanced squeezing regime”, we will add a reference to [33].

2) In nearly all cases even very simple many-body quantum systems have a complicated entanglement Hamiltonian, e.g. free fermions on a 1d chain. From this point of view it is noteworthy and exceptional that the entanglement Hamiltonian of this fully connected spin model is so simple. We will add a sentence to clarify that this is what we have in mind and replace “remarkably” with “noteworthy”.

3) The preprint [33] is already mentioned four times in the manuscript, including twice in the introduction. This is very far from “systematically” omitting its reference. That being said additional references can be added where they were indeed overlooked (see the end of 1) above), also combined with an explanatory sentence beyond the mere citation.

---

## Round 2 · Referee Report · Anonymous (Referee 2) · 2019-12-30

Strengths

This is a nicely written and timely paper studying various aspects of the mean field dynamics incorporating leading Gaussian corrections. The paper analyzes various quenches both within the same phase and across a phase transition (bifurcation point). Several results are obtained analytically and tested against numerics.

Weaknesses

My main issue with the paper, is that many statements are simply overgeneralized without justification. Words "mean field" and "semiclassical" are often mixed together and improperly interchanged. We can have a purely classical model of say colliding macroscopic billiard balls or macroscopic spins - rotators, which under different conditions might or might not be described by Gaussians, mean field regimes etc. It is important to realize that within a purely classical Liouville approach, Gaussian shape of the classical probability distribution is generally not preserved unless the equations of motion are linear or approximately linear. Also the paper several times refers to the truncated Wigner approximation (TWA) as a proper semiclassical limit but no TWA results are shown. I am personally confident that the regime of validity of th esemiclassical TWA for this model applies to much much longer times than the regime of validity of the Gaussian ansatz and many of the time scales mentioned in the paper have little to do with the regime of validity of the semiclassical approach.

Report

Here are more detailed comments.

1) I have immediately trouble with one of the first sentences in the paper: "Another unique feature of quantum mechanics is entanglement, which has no immediate classical analog [1,2]. When entanglement of a composite system is measured by means of the von Neumann entanglement entropy, the logarithm of the Hilbert space dimension of the smaller subsystem is an upper bound on the entanglement. " This statement, while often mentioned is simply incorrect. We can say in the same way that the thermodynamic entropy defined through the logarithm of the density of states is a "unique feature of quantum thermodynamics", which is fo course wrong. Like with any entropy in the classical limit only changes are well defined. And for the quenches considered in this paper or just general quenches, even starting from a pure state, in the appropriate classical limit (say large N) changes of the entanglement entropy will be completely classical, $\hbar$ will drop out from any answer. One can define the entropy of the reduced classical probability distribution even if $\hbar=0$, it is certainly and analogue of entanglement, and deal with it simply subtracting unimportant constant. There are also direct classical analogues like mutual information (which reduces to twice the entanglement for a pure quantum state). Of course, there are quantum regimes like ground states, where everything is quantum, from correlation functions to entanglement. I do not want to start the debate here, but I think the authors should be at least more respectful of classical mechanics and make less categorical statements. There are other similar instances in the paper, which I am not going to highlight.

  1. When the authors derive the MF Hamiltonian (Eq. (2b)) they might want to comment it is integrable, which does not have to be the case. One can take a simple generalization of the model, where spins are split into two groups with N/2 spins in each. Then generally the MF Hamiltonian will be nonintegrable. I believe it is important because for a chaotic MF Hamiltonian the regime of validity of the Gaussian approximation extensively used in the paper (but NOT of the semiclassical approximation) will be even less. I got a general feeling that some of the general conclusions of the paper are actually fine tuned to this very special regime where the large N limit is non-chaotic.

  2. Page 6. "These approximations are believed to be valid as N → ∞." Believed by whom? Is there a Ref.? Actually the validity of semi-classics (TWA) in the large N limit is proven from the path-integral representation. In this seance "are believed" is a bit vague. Similarly one can prove the validity of the Gaussian approximation if the initial conditions are also appropriate. For this model it is easy to write down leading corrections to TWA/Gaussian and show rigorously that they vanish in the large N limit.

  3. Page 7. "It is a non-trivial fact that the time evolution of the variances does not depend on higher moments, such as the skewness, but only on the expectation values. This is a special case of a more general result. Namely, that the dynamics of the leading order of the nth moment depend only on moments of order smaller than n (more details in Appendix B)." I think this statement is not completely untrivial, but interesting of course. For Liouville dynamics for a spin in any time dependent field the Gaussian probability distribution (Wigner function) remains Gaussian. This is a well known fact. Within the Gaussian approximation the MF dynamics is precisely of this nature: spin in some self-consistently determined effective field. This argument can be pushed further to argue that near instabilities like a quench to a Mexican hat (bifurcation point) the Gaussian approximation quickly breaks down. For example, in Ref. https://journals.aps.org/pra/pdf/10.1103/PhysRevA.79.042703, where ramps across a similar singularity were analyzed TWA (justified by large N) perfectly described a crossover from a trivial Gaussian to a nontrivial Gumbel type distribution. At the same time large N Gaussian approximation similar to the one analyzed in this paper, which was studied in a prior work in Ref. Phys. Rev. Lett. 100, 063602 􏰀2008􏰁, had much smaller regime of validity and lead to some spurious results. If there is any lesson from these two papers is that the Gaussian and semiclassical large N limits are completely different (semiclassical includes Gaussian but not the other way around). For this reason it is very important not to mix "semiclassical" and "Gaussian" approximations as it is constantly done in the paper. While formally both are controlled by the same N, but in a very different ways even parametrically. In chaotic MF models it is even more clear, the Gaussian approximation can break down in $Log(N)$ time, while the semiclassical analysis can work for infinitely long times in the appropriate limit.

  4. Typo on page 11: "starts to increases"

  5. Page 14. There are several statements, which are simply incorrect or at least questionable. "A necessary condition for the validity of the saddle point approximation, on which the semiclassical results (4) and (5) rely, is that |ψ|2 in (3) remains localized on a scale of 1/√N. More precisely, the leading order saddle point approximation breaks down when the inverse curvature of the rate function at the saddle point is of the order of the saddle point parameter, i.e. N." We can start from a wave function, which is not even close to a Gaussian, i.e. not localized. Take e.g. an eigenstate of a particle in a square well potential at high energy (level number $n$ is akin to $1/\hbar$). The probability distribution/Wigner function is not a Gaussian by any measure (it is a broadened by $\hbar$ micro canonical ensemble) and yet the saddle point is well defined and Newton's equations will work well in the limit of small $\hbar$/ large $n$. In real time evolution the validity of the semiclassics is NOT controlled by the shape of the distribution. If one uses "sufficient" instead of "necessary" then yes: if the distribution is initially a narrow Gaussian and it remains localized in phase space in time, then indeed one can use semiclassics, but not the other way around. The next statement about the Ehrenfest time is equally unjustified. If we extrapolate it to billiard balls or airplanes we will conclude that semiclassical (or simply classical) dynamics never applies as for chaotic systems the Ehrenfest time is tiny even if $\hbar$ is very small. If we are interested in full details of a microscopic trajectory or in e.g. OTOC/quantum echo then yes, but if we are interested in observables analyzed in the paper very likely including the entanglement then the Ehrenfest time is completely irrelevant to the validity of semiclassics.

  6. I sincerely believe the paper will strongly improve from adding TWA results, which as I mentioned will likely be valid for a much longer times. Doing so would remove (or justify) many erroneous claims. If the authors do not have time now I would strongly request to maximally remove reference to semiclassics replacing it with "Gaussian" approximation. One can differently address the same question. Suppose we have no $\hbar$ and everything is classical described by the same Hamiltonian. We initialize our state in some Gaussian, say thermal, but it does not matter and then ask exactly same questions as in the paper. The width of the Gaussian can be gradually sent to zero again in $\hbar$ independent way. Then the classical evolution is by definition valid for infinite times. But I believe many approximations will break down as discussed in the paper.

To summarize, I really like the paper and I find results to be interesting. I also do not doubt any technical details. My main issues are with their interpretation. Even if thee authors disagree with me, I hope they will consider reducing the level of their claims. I do hope that they can add some TWA analysis to the paper, as it amounts to solving the same equations of motion with some extra averaging over initial conditions. Comparing TWA with the Gaussian ansatz will clarify many points and, I think, will be very useful. If the authors can not add TWA for some reason, they should at least carefully reconsider all their claims and interpretations.

---

## Editorial Decision

awaiting_resubmission